# MARC-3, a membrane-associated ubiquitin ligase, is required for fast polyspermy block in *Caenorhabditis elegans*

Ichiro Kawasaki [1,4], Kenta Sugiura [1,4], Taeko Sasaki [2], Noriyuki Matsuda [3], Miyuki Sato [2,4] ✉ & Ken Sato [1] ✉

In many sexually reproducing organisms, oocytes are fundamentally fertilized with one sperm. In *Caenorhabditis elegans*, chitin layer formation after fertilization by the EGG complex is one of the mechanisms of polyspermy block, but other mechanisms remain unknown. Here, we demonstrate that MARC-3, a membrane-associated RING-CH-type ubiquitin ligase that localizes to the plasma membrane and cortical puncta in oocytes, is involved in fast polyspermy block. During polyspermy, the second sperm entry occurs within approximately 10 s after fertilization in MARC-3-deficient zygotes, whereas it occurs approximately 200 s after fertilization in *egg-3* mutant zygotes defective in the chitin layer formation. MARC-3 also functions in the selective degradation of maternal plasma membrane proteins and the transient accumulation of endosomal lysine 63-linked polyubiquitin after fertilization. The RING-finger domain of MARC-3 is required for its in vitro ubiquitination activity and polyspermy block, suggesting that a ubiquitination-mediated mechanism sequentially regulates fast polyspermy block and maternal membrane protein degradation during the oocyte-to-embryo transition.

In many sexually reproducing animals, the oocyte fuses with only one sperm during fertilization to prevent the fertilized egg from having an aneuploid genome, except for a few animals, including newts and birds[1]. In oviparous animals such as sea urchins and frogs, the egg cell membrane is temporarily depolarized within a few seconds after a fusion between the sperm and oocyte membranes, which prevents polyspermy. This reaction is referred to as a fast block to polyspermy. In many animals, including mammals, a permanent slow block to polyspermy is established by forming a fertilization membrane via the exocytosis of cortical granules, which deliver specific components to change the extracellular environment[2,3]. The formation of the fertilization membrane completely prevents additional sperm entry[2]. However, it is unclear whether a fast block to polyspermy occurs in mammalian eggs because they do not depolarize immediately after

fertilization[3]. In *Caenorhabditis elegans* (*C. elegans*), the plasma membrane (PM) of oocytes is considered to be first tightly surrounded by a vitelline layer. The eggs then form a chitin layer inside the vitelline layer within 5 min of fertilization to prevent polyspermy[4]. Cortical granule exocytosis is not essential for the polyspermy block in *C. elegans*[5]. In this process, CHS-1, a chitin synthase localized in the oocyte PM, is suggested to undergo activation for eggshell chitin synthesis[6]. CHS-1 is thought to form a complex with the transmembrane proteins EGG-1 and EGG-2 and the peripheral membrane proteins EGG-3, EGG-4, and EGG-5[7–9]. Reportedly, depletion of EGG-complex components results in loss of the chitin layer and a certain frequency of polyspermy[9,10]. When *chs-1* was knocked down by RNA interference (RNAi), not all eggs (at most 40%) showed a polyspermy phenotype, and only two or three sperm entered the cytoplasm of the

[1]Laboratory of Molecular Traffic, Institute for Molecular and Cellular Regulation, Gunma University, Maebashi, Gunma 371-8512, Japan. [2]Laboratory of Molecular Membrane Biology, Institute for Molecular and Cellular Regulation, Gunma University, Maebashi, Gunma 371-8512, Japan. [3]Department of Biomolecular Pathogenesis, Tokyo Medical and Dental University, Tokyo 113-8510, Japan. [4]These authors contributed equally: Ichiro Kawasaki, Kenta Sugiura, Miyuki Sato. ✉e-mail: m-sato@gunma-u.ac.jp; sato-ken@gunma-u.ac.jp

polyspermy eggs[10]. These results suggest the existence of additional polyspermy-blocking mechanisms other than chitin layer formation.

During fertilization, a mature oocyte is converted to a zygote and acquires totipotency, which enables its differentiation into all cell types. Fertilized eggs contain maternal components, including proteins, RNAs, and other constituents required for early embryogenesis. The embryo subsequently drives zygotic expression to generate embryonic material as embryogenesis proceeds. This process is referred to as the oocyte-to-embryo (zygote) transition. During this process, a proportion of the maternally supplied components are selectively degraded. For example, the degradation of maternal mRNAs and that of oocyte cytosolic proteins by the ubiquitin-proteasome system (UPS) occur during the oocyte-to-embryo transition[11,12]. In addition, recent studies have revealed that fertilization activates two lysosomal degradation pathways, autophagy and endocytosis, which are required for the specific degradation of organelles and proteins derived from each gamete[13,14]. In mice, autophagy is strongly induced in zygotes after fertilization and is subsequently upregulated at the 4–8 cell embryonic stage for supplying amino acids to drive protein synthesis necessary for embryogenesis[15,16]. In *C. elegans*, sperm-derived paternal mitochondria and membranous organelles (MOs) are selectively eliminated via allophagy (allogeneic organelle autophagy) during early embryogenesis, resulting in maternal inheritance of mitochondrial DNA[17–19]. Maternal PM proteins are also selectively degraded after fertilization. In *C. elegans* zygotes, a subset of maternal PM proteins is selectively endocytosed from the PM and delivered to lysosomes via endosomes for degradation. This process involves ubiquitination[5]. In fertilized eggs, lysine (K)-63-linked ubiquitination is drastically upregulated on endosomes during the second meiotic anaphase[20]. The E2 ubiquitin-conjugating enzyme, UBC-13, and its variant, UEV-1, form a complex and are involved in this ubiquitination. Loss of UBC-13 or UEV-1 results in reduced K63-linked ubiquitin signals on endosomes, inhibiting lysosomal degradation of maternal PM proteins[20]. A similar phenomenon has been observed in mammalian embryos[21]. In mouse embryos, several maternal PM proteins, such as Glyt1a, a glycine transporter, are internalized from the PM at the late 2-cell stage and delivered to lysosomes for degradation at the 4-8-cell stage[22]. Strong ubiquitination signals are observed on the endosomes during this process. Although the selective degradation of maternal PM proteins is a conserved process in *C. elegans* and mammals, the molecular mechanisms involved and their physiological roles during development remain largely unknown.

In this study, we demonstrate the involvement of the ubiquitin ligase MARC-3 and E2 ubiquitin-conjugating enzyme LET-70 in selective maternal PM protein degradation in *C. elegans*. Our results show that MARC-3 is strongly expressed in growing oocytes and localized to the PM and punctate structures. MARC-3 is internalized from the PM during oocyte maturation and ovulation, and is subsequently delivered to lysosomes for degradation during early embryogenesis. Furthermore, we reveal that loss of MARC-3 leads to a polyspermy phenotype during the early phase of fertilization, suggesting that a ubiquitination-mediated mechanism is involved in the fast polyspermy block that occurs immediately after the contact of the first sperm. These results suggest that MARC-3 has multiple functions, including fast polyspermy block upon fertilization and selective maternal membrane protein degradation during early development.

## Results

### MARC-3 is involved in maternal membrane protein degradation

In *C. elegans*, various maternal PM proteins are internalized and delivered to lysosomes for degradation after fertilization. Previously, we demonstrated that the clearance of maternal PM proteins and the accumulation of K63-linked polyubiquitin signals on endosomes depend on UBC-13 and UEV-1, which form an E2 complex mediating K63-linked polyubiquitination[20]. To identify the E3 ubiquitin-protein ligase that functions in the degradation of maternal PM proteins[23], we focused on *C. elegans* homologs of mammalian Membrane-Associated Ring finger C3HC4 (MARCH) E3 ubiquitin ligases, each of which is localized to the membrane of specific organelle. In *C. elegans*, there are six homologs of the MARCH protein family (MARC-1–6) (Supplementary Fig. 1). We performed RNAi knockdown of these genes and observed that loss of MARC-3 delayed the degradation of some maternal PM proteins (Fig. 1 and Supplementary Fig. 2).

MARC-3 is the *C. elegans* homolog of mammalian MARCH3 and contains a RING-CH domain and two putative transmembrane domains (Fig. 1A and Supplementary Fig. 1). According to a previous gene expression profiling study, *marc-3* is the only member that is highly expressed in the adult hermaphrodite germline among the *marc* family members[24]. Mammalian MARCH3 localizes to endosomes and participates in transferrin uptake[25]. To elucidate the function of MARC-3, we used either RNAi or a deletion mutant, *marc-3(tm1626)*, which lacks a 520-bp genomic sequence including the promoter region and a coding region encompassing most of the RING-CH domain of MARC-3 (Fig. 1A).

We first examined whether the degradation of GFP-tagged maternal membrane proteins was affected in *marc-3(tm1626)* mutant hermaphrodites. Adult *C. elegans* hermaphrodites have a pair of U-shaped tubular ovaries connected to the uterus via the spermatheca (Fig. 1B). The most proximal oocyte adjacent to the spermatheca (the −1 oocyte) undergoes meiotic maturation and cortical rearrangement and is subsequently ovulated into the spermatheca, where fertilization occurs. After fertilization, the zygote completes meiosis I and II and initiates embryonic development in the uterus. In adult wild-type (WT) hermaphrodites, a subset of maternal membrane proteins, including CAV-1, RME-2, and CHS-1, were promptly endocytosed and degraded after fertilization, and their signals were significantly diminished after the 2-cell embryonic stage (Fig. 1C, D and Supplementary Fig. 2). When CAV-1 was tagged with GFP at its C-terminus (CAV-1::GFP) and expressed under the control of a germline-specific *pie-1* promoter, it localized mainly to the cortical granules and PM in growing oocytes (Fig. 1C). After fertilization, CAV-1::GFP is targeted to the PM by cortical granule exocytosis and is then readily delivered to lysosomes for degradation[5,26]. RME-2 is a yolk receptor that recycles between the PM and endosomes to take up yolk components from the body cavity[27]. GFP-tagged RME-2 accumulated in the endosomes of maturing oocytes just before fertilization and was targeted to lysosomes for degradation (Supplementary Fig. 2A). CHS-1 is a chitin synthase essential for eggshell chitin layer formation and its GFP-fusion protein mainly localized on oocyte PM and was delivered to lysosomes for degradation after fertilization (Supplementary Fig. 2B). The degradation of these GFP-tagged maternal membrane proteins was delayed in adult *marc-3(tm1626)* mutant hermaphrodites compared to WT controls (Fig. 1C, D and Supplementary Fig. 2). Quantification of GFP signal intensities in WT and *marc-3(tm1626)* mutant oocytes and embryos confirmed these observations. The signal intensities of these maternal membrane proteins were not significantly different between WT and *marc-3(tm1626)* mutant oocytes but were higher in *marc-3(tm1626)* mutant embryos than in WT embryos (Fig. 1D and Supplementary Fig. 2). Thus, we conclude that MARC-3 is involved in the degradation of a subset of maternal membrane proteins, possibly as an E3 ubiquitin ligase.

We examined whether other members of the MARC protein family were involved in this degradation process. Among the MARC family proteins, MARC-2 was most similar to MARC-3 (Supplementary Fig. 1). Knockdown of each family member by RNAi showed that the loss of any *marc* genes, except *marc-3*, did not significantly affect CAV-1::GFP degradation in the WT genetic background (Fig. 1E). Furthermore, knockdown of each family member via RNAi in a *marc-3(tm1626)* mutant background did not further increase the undegraded CAV-1::GFP signal compared with the *marc-3(tm1626)* mutant (Fig. 1F). These

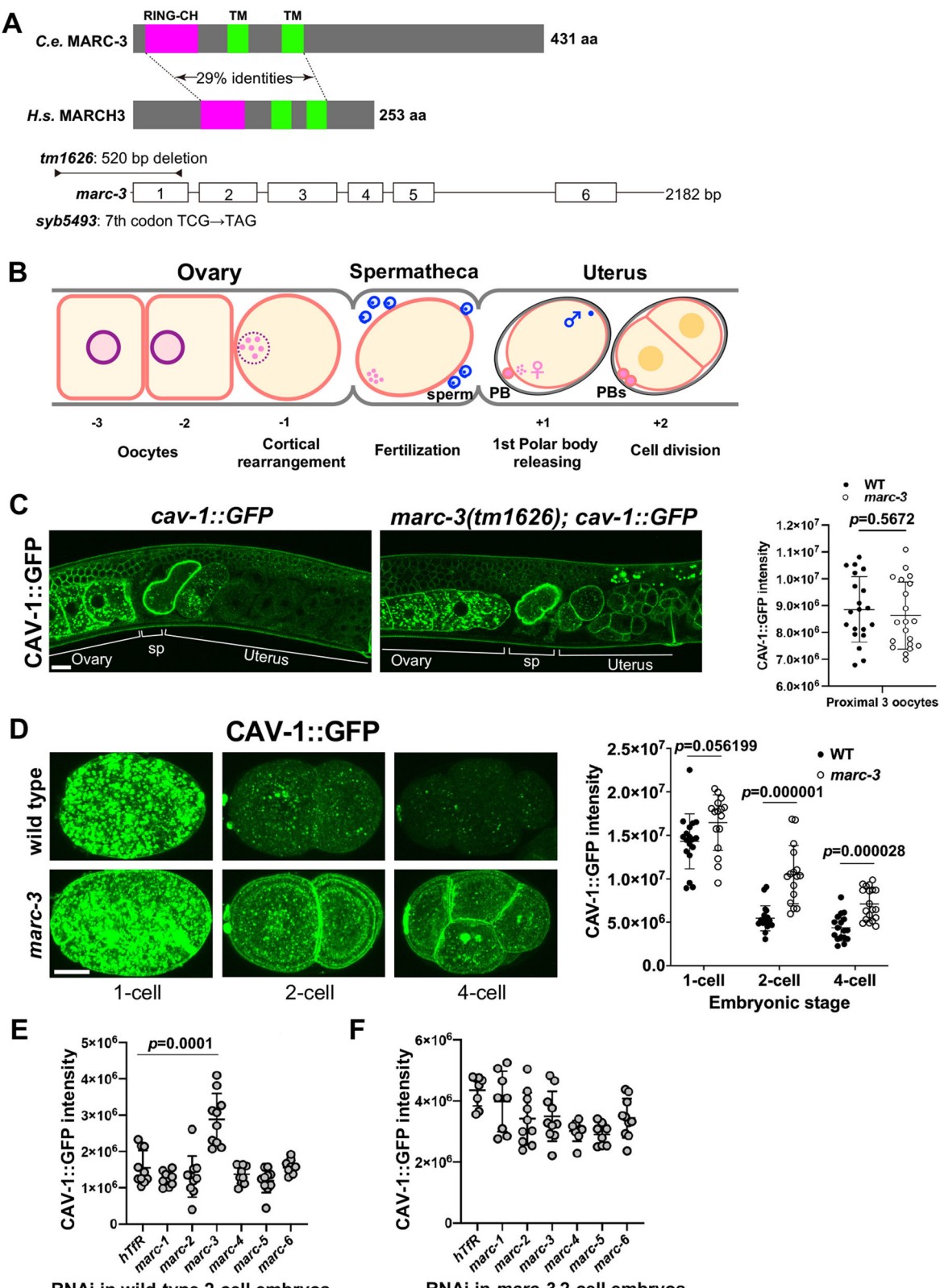

results suggest that among the six MARC family members, MARC-3 is predominantly involved in the degradation of maternal PM proteins during early embryogenesis.

## K63-linked polyubiquitin is reduced in *marc-3* mutant zygotes

Fertilization induces the transient accumulation of K63-linked ubiquitin chains on endosomes of *C. elegans* zygotes. We previously reported that the K63-linked ubiquitination activity was significantly reduced in *ubc-13* and *uev-1* mutants[20]. To examine whether MARC-3 is involved in this ubiquitination process, we performed immunostaining of zygotes of various mutants expressing CAV-1::GFP using an anti-ubiquitin antibody, which recognizes protein-conjugated, but not free, ubiquitin (FK2) (Fig. 2). In the middle-to-late meiosis II-stage zygotes of WT and *marc-2(syb3694)* mutants, CAV-1::GFP signals were mostly

**Fig. 1 | MARC-3 is required for the degradation of a subset of maternal plasma membrane proteins. A** Top, *C. elegans* MARC-3 and *Homo sapiens* MARCH3, the two homologs containing a RING-CH domain (magenta) and two transmembrane domains (green). Bottom, *marc-3* unspliced transcript and the two mutations examined. Boxes, exons. Lines, UTRs and introns. *tm1626* is a 520-bp deletion encompassing most of the RING-CH domain. *syb5493* introduces a premature termination at the 7th codon. **B** *C. elegans* adult hermaphrodite gonad proximal region containing oocytes and embryos in the ovary, spermatheca, and uterus. PB, polar body. **C** Degradation of CAV-1::GFP is delayed in *marc-3(tm1626)* adult hermaphrodites (right) compared with wild-type control (left). Bar, 10 μm. SP, spermatheca. Right, distribution of CAV-1::GFP intensities obtained from middle-focal-plane images of proximal three oocytes in wild-type (black) and *marc-3(tm1626)* (white) adult hermaphrodites. *n* = 20. *p* = 0.5672. **D** CAV-1::GFP signals in *z*-stack images of wild-type and *marc-3(tm1626)* zygotes, 2-, and 4-cell embryos. Left, representative *z*-stack images of wild-type and *marc-3(tm1626)* embryos at respective stages. Bar, 10 μm. Right, distribution of CAV-1::GFP intensities per embryo at three embryonic stages. *n* = 18, 19, and 17 for wild type (black), *n* = 16, 16, and 19 for *marc-3(tm1626)* (white), and *p* = 0.056199, 0.000001, and 0.000028 at the 1-, 2-, and 4-cell stages, respectively. **E** Distribution of CAV-1::GFP intensities per wild-type 2-cell stage embryo, where each *marc*-family gene was RNAi depleted. *n* = 9 embryos for *marc-1, -4,* and *-6* RNAi. *n* = 10 for *hTfR, marc-2, -3,* and *-5* RNAi. *hTfR* RNAi vs. *marc-3* RNAi, *p* = 0.0001. **F** Distribution of CAV-1::GFP intensities per *marc-3(tm1626)* 2-cell stage embryo, where each *marc*-family gene was RNAi depleted. *n* = 10 for each RNAi. Y axis, arbitrary units. *hTfR* RNAi, a negative control. *p* values were calculated using multiple unpaired *t*-tests (two-tailed). Horizontal lines, mean ± S.D.

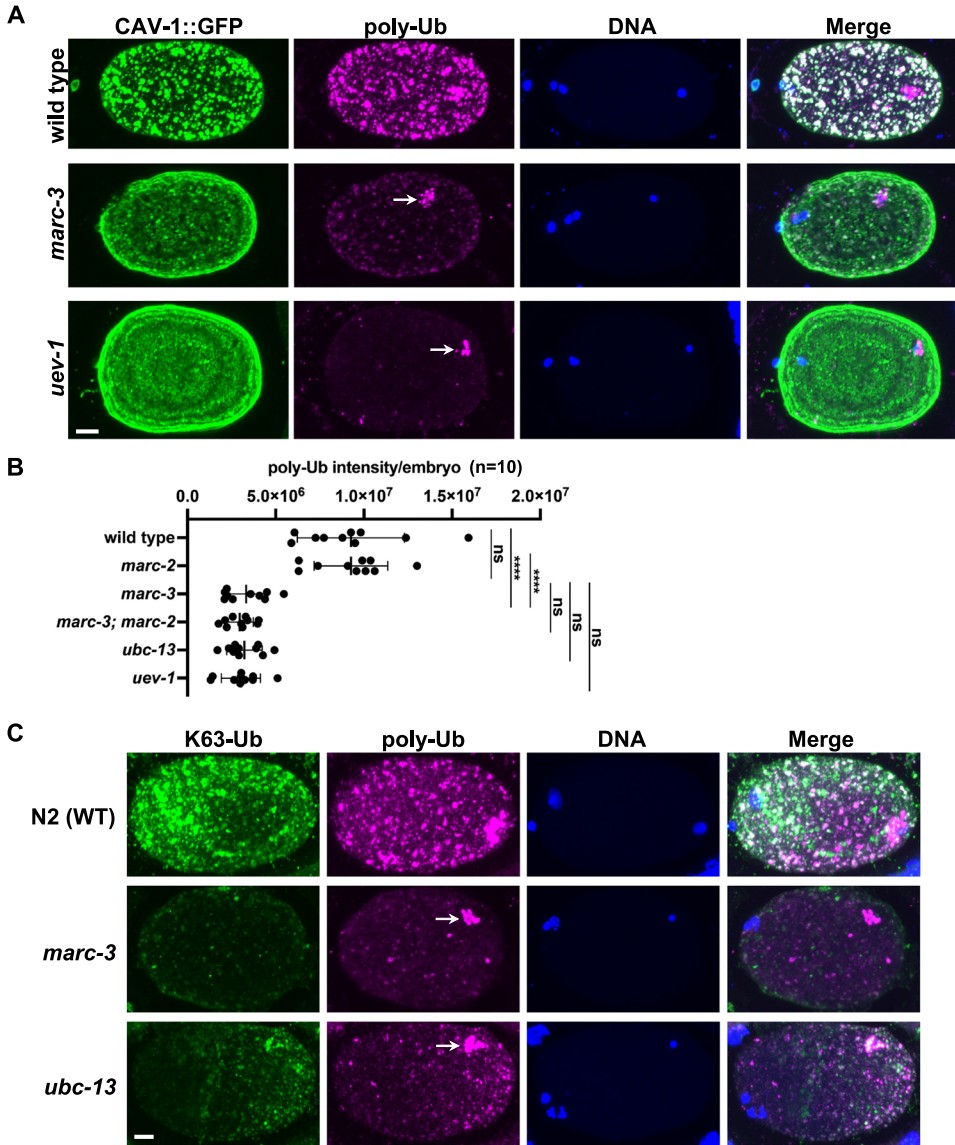

**Fig. 2 | Loss of MARC-3 reduces endosome-associated K63-linked polyubiquitination in zygotes. A** Immunostaining of zygotes of WT and various mutants expressing CAV-1::GFP (green) with an anti-ubiquitin antibody, which recognizes protein-conjugated ubiquitin and not free ubiquitin (FK2, magenta). DNA staining with DAPI, blue. Arrows indicate ubiquitination on MOs. Bar, 5 μm. Observed embryos, *n* = 8 for WT, *n* = 18 for *marc-3*, and *n* = 10 for *uev-1*. **B** Distribution of ubiquitin signal intensities per zygote of WT and various mutants. Quantified embryos, *n* = 10 for each genotype. Ubiquitin signal intensities were digitally quantified as in Fig. 1 and are shown in arbitrary units. Statistical significance was calculated using multiple unpaired *t*-tests (two-tailed). Obtained *p* values were as follows: wild type vs. *marc-2*; *p* = 0.9967, ns (not significant). wild type vs. *marc-3*; ****p* < 0.0001. *marc-2* vs. *marc-3*; ****p* < 0.0001. *marc-3* vs. *marc-3; marc-2*; *p* = 0.4330, ns. *marc-3* vs. *ubc-13*; *p* = 0.8422, ns. *marc-3* vs. *uev-1*; *p* = 0.5674, ns. Vertical lines in the graph indicate mean ± S.D. **C** Double immunostaining of zygotes of wild-type N2, *marc-3(tm1626)*, and *ubc-13(tm3546)* with anti-K63-linked polyubiquitin antibody (Apu3, green) and anti-polyubiquitin antibody (FK2, magenta). DNA staining, blue. Arrows indicate ubiquitination on MOs. Bar, 5 μm. Observed embryos, *n* = 10 for each genotype. See Supplementary Fig. 3.

detected on endosomes, in which protein-conjugated ubiquitin signals accumulated prominently (Fig. 2A, B). Meanwhile, in the zygotes of *marc-3(tm1626)* single and *marc-3(tm1626); marc-2(syb3694)* double mutants, a significant proportion of CAV-1::GFP signals remained on the PM, and the levels of ubiquitin signals on endosomes were significantly reduced compared with those in WT and *marc-2(syb3694)* zygotes, similar to the *ubc-13(tm3546)* and *uev-1(ok2610)* zygotes (Fig. 2A, B). Sperm-derived MOs are ubiquitinated immediately after fertilization and detected via immunostaining with anti-polyubiquitin antibodies[17,28]. In contrast with ubiquitination on endosomes, MOs around paternal pronuclei remained ubiquitinated in *marc-3(tm1626)* zygotes and in *uev-1(ok2610)* zygotes (Fig. 2A, arrows).

To determine the type of polyubiquitin chain affected in *marc-3(tm1626)* mutants, we performed immunostaining of N2 (WT), *marc-3(tm1626)*, and *ubc-13(tm3546)* zygotes with anti-K48- and anti-K63-linked polyubiquitin antibodies (Apu2 and Apu3, respectively) (Fig. 2C and Supplementary Fig. 3)[29]. We observed that anti-K48-polyubiquitin immunostaining, which showed strong signals on MOs around paternal pronuclei in WT zygotes, was not grossly changed in *marc-3(tm1626)* and *ubc-13(tm3546)* mutant zygotes (Supplementary Fig. 3). In contrast, anti-K63-polyubiquitin immunostaining, which mainly showed signals on the endosomes in WT zygotes, was significantly reduced in *marc-3(tm1626)* and *ubc-13(tm3546)* mutant zygotes (Fig. 2C). These results indicated that MARC-3 is required for K63-linked polyubiquitination in zygotes, like UBC-13.

## MARC-3 shows dynamic behaviors in oocytes and early embryos

Next, we examined the subcellular localization of MARC-3 in oocytes, zygotes, and embryos. We generated a transgenic strain expressing MARC-3 tagged with GFP at its C-terminus (MARC-3::GFP) under the control of the *pie-1* promoter (Fig. 3 and Supplementary Fig. 4A). This *marc-3::gfp* transgene complemented the endosomal ubiquitination defect of *marc-3(tm1626)* mutant zygotes, indicating that MARC-3::GFP fusion protein is functional (Supplementary Fig. 4B). In a cell fractionation experiment, MARC-3::GFP was fractionated into the membrane fraction, indicating that it is a transmembrane protein (Fig. 3A). We next compared the subcellular localization of MARC-3::GFP with that of the small GTPases RAB-5 and RAB-7, which are localized to the early and late endosomes, respectively (Fig. 3B, C and Supplementary Fig. 5, Movie 1). In growing oocytes, MARC-3::GFP was predominantly localized to the PM and cortical punctate structures, which partially overlapped with mCherry-tagged RAB-5 and mRFP-tagged RAB-7, indicating that MARC-3 was localized to the PM and endosomes (Fig. 3B and Supplementary Fig. 5). MARC-3::GFP was internalized from the PM, accumulated on enlarged mCherry::RAB-5-positive early endosomes in ovulated oocytes and fertilized zygotes, and was degraded during early embryogenesis (Fig. 3B, C). MARC-3::GFP was also detected in mRFP::RAB-7-positive late endosomes in zygotes, then, disappeared in later-stage embryos (Supplementary Fig. 5B). We also generated a transgenic strain expressing C-terminally GFP-tagged MARC-3 endogenously by inserting a DNA fragment encoding GFP before the stop codon of *marc-3* using the CRISPR-Cas9 genome-editing system. We confirmed that endogenous MARC-3::GFP expressed under its own promoter showed a very similar localization pattern as *pie-1* promotor-driven MARC-3::GFP (Supplementary Fig. 6A). We further generated an anti-MARC-3 antibody and examined the localization of endogenous MARC-3 proteins during the oocyte-to-embryo transition by immunostaining with the antibody (Supplementary Fig. 6B–E). Endogenous MARC-3 was detected in growing oocytes starting from early diakinesis (Supplementary Fig. 6B). MARC-3 was localized to the PM and punctate structures under the PM in late-stage oocytes (Supplementary Fig. 6B, D) and was eventually internalized from the PM and degraded in embryos after fertilization (Supplementary Fig. 6C–E). Endogenous MARC-3 as well as MARC-3::GFP were also detected on internal punctate structures, presumably the Golgi

apparatus. These results suggest that the subcellular localization of MARC-3 is dynamically regulated during oogenesis, and that MARC-3 is a maternal substrate that is endocytosed in ovulated oocytes and degraded immediately after fertilization.

## MARC-3 binds to LET-70 and shows ubiquitin ligase activity

A ubiquitin-conjugating enzyme (E2) is brought to its substrate by binding to a ubiquitin protein ligase (E3) that binds to E2 and its substrate. Because both MARC-3, a predicted E3 ligase, and the UBC-13/UEV-1 E2 complex were shown to function in K63-linked ubiquitination in zygotes, we examined whether they interact physically to work together as an E2-E3 complex using yeast two-hybrid analysis (Fig. 4A). We used the N-terminal cytoplasmic region of MARC-3 (98 aa), which contains the RING-CH-type domain, as the bait. When we tested either full-length UBC-13 or UEV-1 as prey, we could not detect an activation signal for the reporter gene (*LacZ*) (Fig. 4A). We investigated other E2 ubiquitin-conjugating enzymes as partners of MARC-3 in the degradation of maternal PM proteins[20]. Among the E2 enzymes in *C. elegans*, LET-70 is involved in essential ubiquitination-related biological processes during the oocyte-to-embryo transition[30]. Thus, we examined the interaction of MARC-3 with full-length LET-70 using yeast two-hybrid analysis and observed that the N-terminal MARC-3 region binds to LET-70 (Fig. 4A).

Next, we examined whether LET-70 was involved in maternal PM protein degradation. However, loss of LET-70 is known to cause a strong lethal phenotype[30]. Therefore, we treated hermaphrodites expressing CAV-1::GFP or GFP::CHS-1 with *let-70* RNAi for a limited period, from the L4 larval stage to the young adult stage and examined the effects (Fig. 4B). We observed that the degradation of CAV-1::GFP and GFP::CHS-1 in early embryos was significantly delayed in *let-70* RNAi-treated hermaphrodites (Fig. 4B), like that in *marc-3(tm1626)* hermaphrodites (Fig. 1C, D, and Supplementary Fig. 2). These results suggest that LET-70 is involved in the degradation of maternal PM proteins in early embryos.

To further examine whether LET-70 and MARC-3 could function as an active E2-E3 complex, we monitored the ubiquitin ligase activity of the MARC-3 RING-CH domain in the presence of UbcH5a, a human E2 ortholog of LET-70, using an in vitro ubiquitination assay system (Fig. 4C). UbcH5a is 94% identical to LET-70 in its amino acid sequence. We observed that the WT MARC-3 RING-CH domain fused to maltose-binding protein (MBP) showed auto-ubiquitination activity in the presence of UbcH5a (Fig. 4C). In contrast, a similar fusion protein construct, in which one of the conserved cysteine residues in the MARC-3 RING-CH domain was converted to serine (C38S), failed to auto-ubiquitinate (Fig. 4C). These results confirmed that the RING-CH domain of MARC-3 is critical for ubiquitination.

## Loss of MARC-3 leads to a polyspermy phenotype

To further investigate the biological significance of MARC-3, we examined additional phenotypes of the *marc-3* mutant. The *marc-3(tm1626)* mutant animals were mostly viable and fertile, but showed a reduced brood size compared with the wild-type N2 animals (Supplementary Table 1; mean ± S.D. = 248.1 ± 18.1, *n* = 10; 80.7% of N2). In addition, 11% of the laid eggs were arrested during embryogenesis and 3.5% were also arrested after hatching (Supplementary Table 1). Furthermore, we observed that some of the *marc-3*-deficient zygotes showed a polyspermy phenotype. Blocking polyspermy is observed in many sexually reproducing animals, including *C. elegans*[4,9,10]. To distinguish between paternal and maternal pronuclei, we immunostained zygotes with the monoclonal antibody 1CB4, which recognizes sperm-derived organelles, MOs, along with DAPI nuclear staining. We detected only one sperm-derived paternal pronucleus in the WT zygotes (Fig. 5A and Table 1, *n* = 56 at 20 °C, *n* = 53 at 25 °C). In contrast to the WT zygotes, we observed that up to 30% of *marc-3(tm1626)* zygotes showed a polyspermy phenotype (Fig. 5B and Table 1, 15 out of 50 at

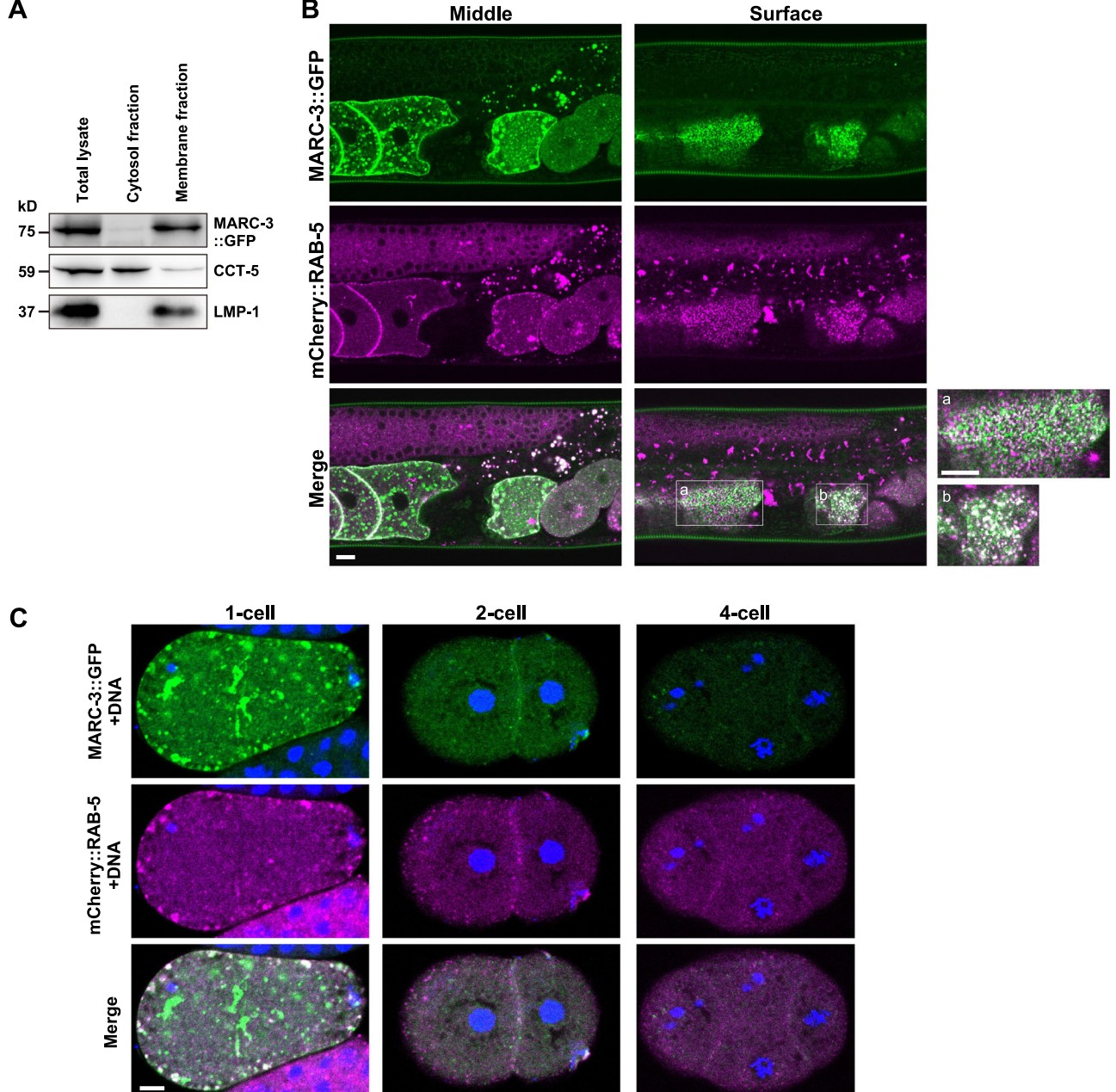

**Fig. 3 | Subcellular localization of MARC-3 in oocytes and embryos compared with RAB-5. A** MARC-3 is a transmembrane protein. The lysate of transgenic animals expressing MARC-3::GFP under the *pie-1* promoter control was fractionated and examined by immunoblotting using specific antibodies. CCT-5 and LMP-1 were used as controls for cytosolic and membrane proteins, respectively. MARC-3::GFP and LMP-1 were fractionated to the membrane fraction. Experiments were repeated independently three times with similar results. **B** Subcellular localization of MARC-3::GFP (green) compared with mCherry::RAB-5 (magenta), an early endosome marker, in growing oocytes and early embryos in an adult hermaphrodite gonad. Both middle and surface focal plane images are shown. a and b, enlarged images of boxed areas. Bars, 10 μm. MARC-3::GFP partially overlapped with mCherry::RAB-5 in ovulated oocytes (*n* = 43 hermaphrodites). **C** Subcellular localization of MARC-3::GFP (green) compared with mCherry::RAB-5 (magenta) in zygotes, 2, and 4-cell-stage embryos. Blue, DNA staining. Bar, 5 μm. MARC-3::GFP was accumulated on mCherry::RAB-5-positive early endosomes in fertilized zygotes (1-cell embryos) and degraded during early embryogenesis (*n* = 22 for zygotes, *n* = 7 for 2-cell embryos, and *n* = 4 for 4-cell embryos). See Supplementary Fig. 5.

20 °C). Most of *marc-3*-mutant polyspermy zygotes contained two paternal pronuclei; however, we observed a few polyspermy zygotes with more than two paternal pronuclei (Fig. 5B, C). To determine whether the polyspermy phenotype is caused by a defect in the sperm or oocytes, we examined the effect of reciprocal matings on the polyspermy phenotype as follows: First, when *marc-3* mutant hermaphrodites were mated with WT males, polyspermy zygotes were still produced (Table 2, 13%, 10 out of 78); Second, when WT hermaphrodites were mated with *marc-3(tm1626)* mutant males,

polyspermy zygotes were not produced (Table 2, 0%, 0 out of 45). These results suggested that oocyte defects caused the polyspermy phenotype of the *marc-3(tm1626)* mutant. To confirm that the loss of *marc-3* function was the cause of the polyspermy phenotype, we generated another loss-of-function *marc-3* allele, *syb5493*, in which a premature termination codon was introduced at the 7th codon position using CRISPR-Cas9 genome editing (Fig. 1A). We observed a polyspermy phenotype in *marc-3(syb5493)* mutant zygotes, like that in the *marc-3(tm1626)* mutant zygotes (Table 1, 18%, 20 out of 111).

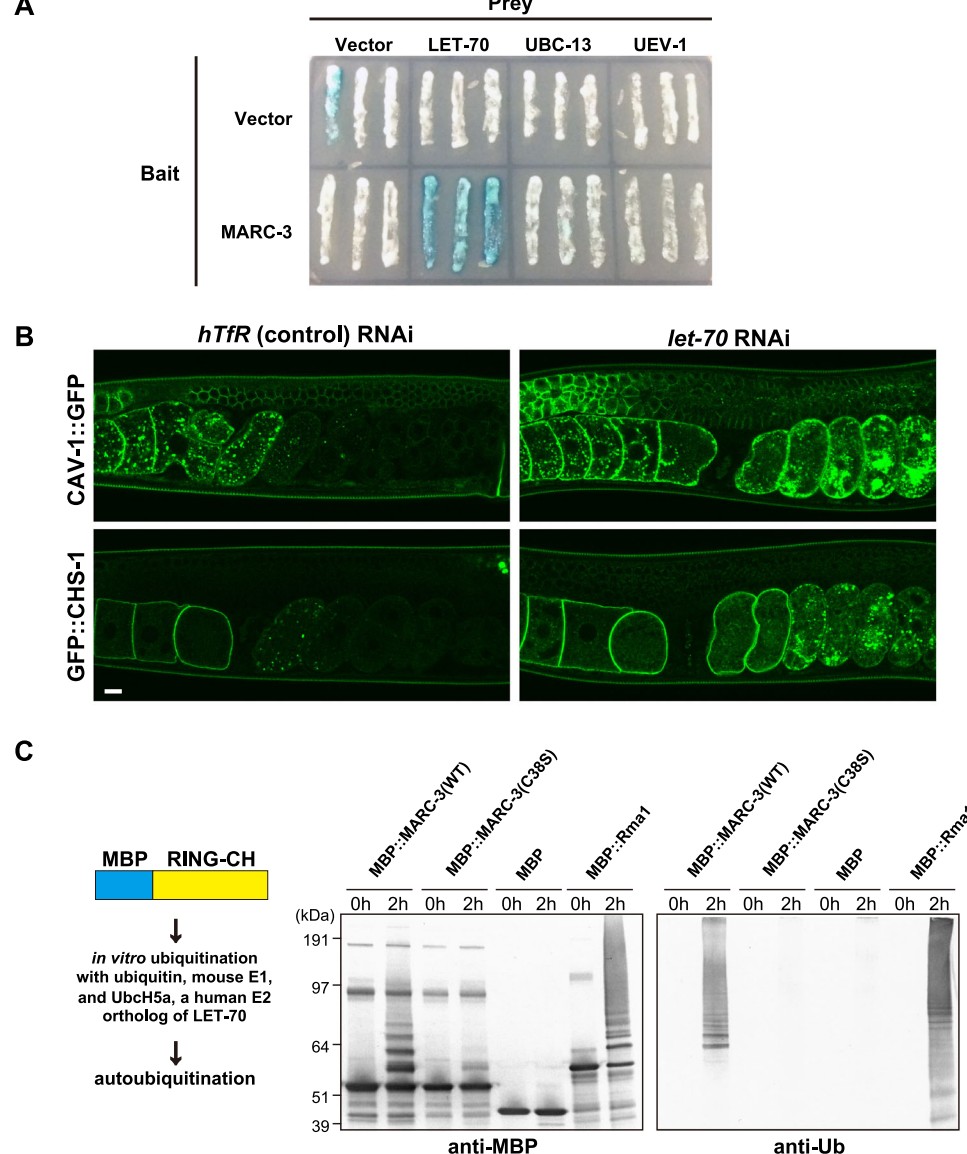

**Fig. 4 | MARC-3 interacts with the ubiquitin-conjugating enzyme LET-70 and shows ubiquitin ligase activity in vitro. A** Yeast two-hybrid analysis testing for interactions between MARC-3 and several E2 ubiquitin-conjugating enzymes. The N-terminal cytosolic region (1–98 aa) of MARC-3, which contains the RING-CH domain, was used as the bait. The full-length LET-70, UBC-13, and UEV-1 were used as prey. LacZ was used as the reporter. The N-terminal region of MARC-3 bound to LET-70 but not to UBC-13 or UEV-1. **B** Degradation of CAV-1::GFP and GFP::CHS-1 after fertilization in adult hermaphrodite gonads was delayed after *let-70* RNAi knockdown (right) compared with *hTfR* negative-control RNAi (left) (*n* = 8 for *hTfR* RNAi on CAV-1::GFP gonads, *n* = 8 for *let-70* RNAi on CAV-1::GFP gonads, *n* = 8 for *hTfR* RNAi on GFP::CHS-1 gonads, and *n* = 16 for *let-70* RNAi on GFP::CHS-1 gonads).

Bar, 10 μm. **C** Maltose-binding protein (MBP), MBP-fused WT MARC-3 RING-CH domain (MBP::MARC-3(WT)), MBP-fused mutated MARC-3 RING-CH domain, in which the 38th cysteine was substituted with serine (MBP::MARC-3(C38S)), and MBP-fused Rma1 (MBP::Rma1), a positive control, were tested for their auto-ubiquitination activities in the presence of UbcH5a, a human E2 ortholog of LET-70. Left column, the scheme for the in vitro ubiquitination assay. Middle column, aliquots of respective reaction mixtures were western blotted and probed with an anti-MBP antibody. Right column, aliquots of reaction mixtures were western blotted and probed with an anti-ubiquitin antibody. Reaction mixtures were incubated for either 0 or 2 h before western blotting. Experiments were repeated independently two times with similar results.

Furthermore, when a transgene encoding functional MARC-3::GFP was introduced into the *marc-3(tm1626)* mutant (Supplementary Fig. 4), the resulting transgenic animals did not exhibit the polyspermy phenotype (Table 2, also see Fig. 7C). Finally, when *marc-3* was RNAi-depleted in the *rrf-1(pk1417)* strain, in which RNAi is predominantly effective in the germline[31], for two consecutive generations, the polyspermy phenotype was successfully phenocopied (Table 1). The results of these genetic analyses indicated that *marc-3* is the causative gene for the polyspermy phenotype. In addition, we examined whether any other genes in the ubiquitin conjugation pathway showed a polyspermy phenotype when mutated or when RNAi was treated. Among the tested genes, which included *marc-2*, *ubc-13*, *uev-1*, *ubc-18*, and *let-70*, only *let-*

*70* RNAi resulted in a polyspermy phenotype albeit less frequently (Table 1; 2.4%, 5 out of 205 zygotes). These results suggest that among the ubiquitin conjugation pathway genes examined, *marc-3* and *let-70* exert a unique function in blocking polyspermy.

## Chitin layer formation occurs in *marc-3* mutant embryos

Previously, several proteins involved in eggshell chitin synthesis, such as GNA-2 and CHS-1, the components of the EGG complex, EGG-1, 2, 3, 4, and 5, and a chitin-interacting protein CBD-1 were reported to be required for polyspermy block in *C. elegans*[9,10]. After RNAi knockdown of these genes, besides showing a polyspermy phenotype, the zygotes either partially or entirely lacked the eggshell chitin layer[8–10].

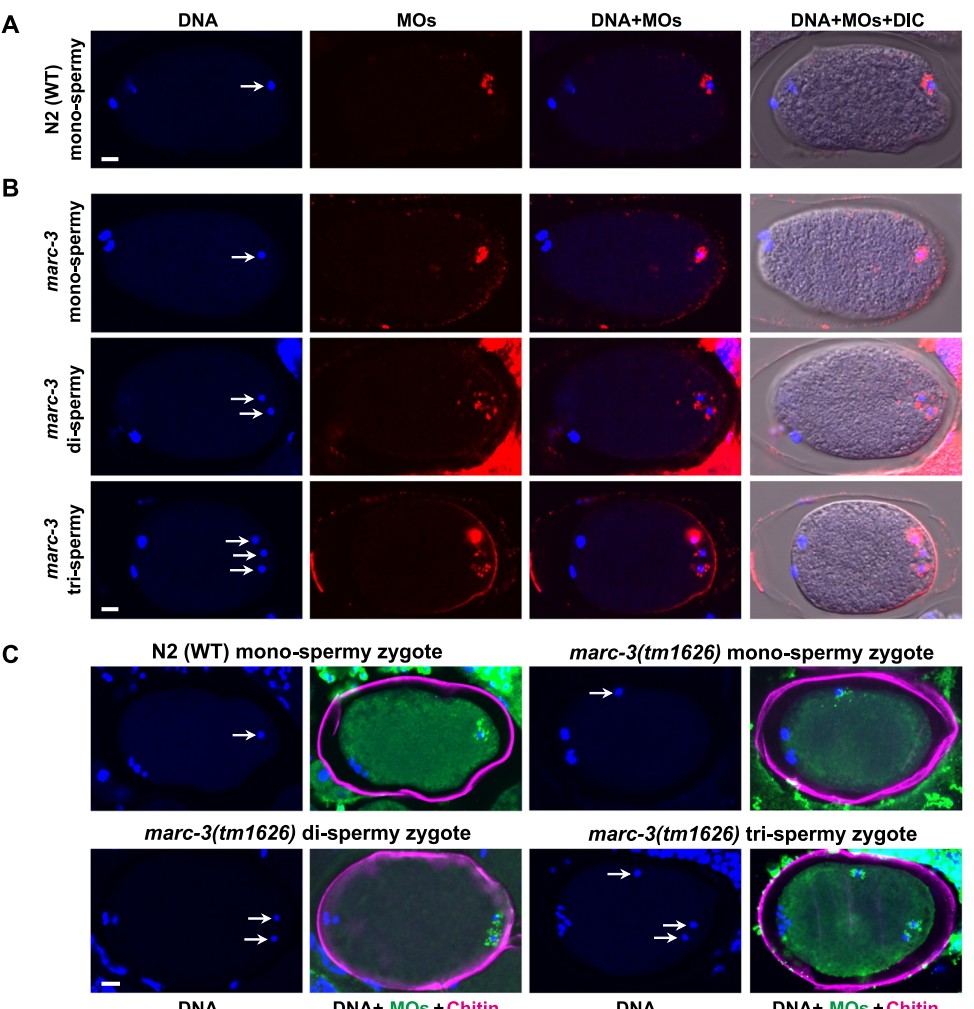

**Fig. 5 | MARC-3-deficient zygotes show a polyspermy phenotype. A** Only one paternal pronucleus was observed in WT zygotes (*n* = 56 zygotes at 20 °C, *n* = 53 at 25 °C). **B** Some of *marc-3(tm1626)* zygotes showed a polyspermy phenotype (15 out of 50 at 20 °C, 8 out of 35 at 25 °C). In (**A**) and (**B**), zygotes were immunostained with a monoclonal antibody, 1CB4, which recognizes sperm-derived MOs (red), a paternal pronuclei marker, with nuclear DNA counterstaining (blue). DIC, differential interference contrast. Arrows indicate paternal pronuclei. Bars, 5 μm.

**C** Immunostaining of N2 (WT) monospermic zygote and *marc-3(tm1626)* monospermic and polyspermic zygotes with a rhodamine-conjugated chitin-binding probe (magenta) together with an anti-MO antibody, 1CB4 (green), and DAPI nuclear staining (blue). Arrows indicate paternal pronuclei. Bar, 5 μm. Like the WT zygotes (*n* = 17), a rhodamine-positive chitin layer was formed around the circumference of *marc-3(tm1626)* zygotes, regardless of whether the zygote was monospermic (*n* = 17) or polyspermic (*n* = 8). See Supplementary Fig. 7.

Furthermore, *spe-11* mutant zygotes, which display a polyspermy phenotype, contained only a patch of the chitin layer[10]. To examine whether *marc-3(tm1626)* zygotes possess an eggshell chitin layer, we immunostained WT and *marc-3(tm1626)* zygotes with the anti-MO antibody 1CB4 and counterstained them with DAPI and a rhodamine-conjugated chitin-binding probe (New England Biolabs) to detect chitin[32] (Fig. 5C). We observed that, like the WT zygotes, a rhodamine-positive chitin layer was formed around the circumference of *marc-3(tm1626)* zygotes, regardless of whether the zygote was monospermic or polyspermic (Fig. 5C). To test whether the chitin layer formed around the *marc-3(tm1626)* mutant zygotes is intact and functional, we exposed unfixed embryos of WT, *spe-11(hc77)* mutant grown at the non-permissive 25 °C, and *marc-3(tm1626)* mutant to a solution containing the lipophilic dye FM4-64 and DAPI (Supplementary Fig. 7). Among them, the *spe-11* mutant embryos grown at 25 °C were permeable to FM4-64 and DAPI. In contrast, the *marc-3* mutant embryos were impermeable to these dyes, except for the embryos exposed to these dyes before the completion of metaphase I, which were permeable to these dyes, as previously described for WT embryos (Supplementary Fig. 7)[26,33,34]. These results suggest that a functional eggshell chitin layer is formed around *marc-3(tm1626)*

mutant embryos. However, the exact timing of its synthesis during development has not been precisely determined. This phenotype is unique compared with that of previously identified polyspermy mutants[9,10].

## MARC-3 functions in the fast polyspermy block

To investigate the relationship between MARC-3 and the EGG-complex components in the regulation of polyspermy block, we compared the phenotypes of the *marc-3* single mutant with those of *egg-3* single mutant, which exhibits defects in the chitin layer formation and the polyspermy block, and *marc-3; egg-3* double mutant. For this purpose, we constructed a *marc-3(tm1626); egg-3(tm1191)/mIn1[mIs14 dpy-10]* balanced double-mutant strain and compared the polyspermy frequencies between its *marc-3; egg-3* double homozygous mutant progeny and the *marc-3; egg-3/mIn1* single homozygous mutant progeny (*marc-3* homo and *egg-3* hetero), along with the polyspermy frequencies of *egg-3* single homozygous mutant progeny and *egg-3/mIn1* heterozygous mutant progeny produced from an *egg-3(tm1191)/mIn1[mIs14 dpy-10]* balanced *egg-3* mutant strain (Table 2). Because the *egg-3* homozygous mutant embryos also display a polarity- and osmotic-sensitivity-defective (Pod) phenotype, which obscures the

**Table 1 | Polysperm phenotype observed in various mutants and after RNAi treatments**

**20 °C**

| Strain | Monosperm | Polysperm | % Polysperm |
|---|---|---|---|
| N2 | 56 | 0 | 0 (0/56) |
| marc-3(tm1626) I | 35 | 15 | 30 (15/50) |
| marc-3(syb5493) I | 91 | 20 | 18 (20/111) |
| marc-2(syb3694) II | 74 | 0 | 0 (0/74) |
| ubc-13(tm3546) IV | 79 | 0 | 0 (0/79) |
| uev-1(ok2610) I | 27 | 0 | 0 (0/27) |
| ubc-18(tm5426) III | 64 | 0 | 0 (0/64) |
| marc-3(tm1626) I; ubc-13(tm3546) IV | 47 | 25 | 35 (25/72) |

**25 °C**

| Strain | Monosperm | Polysperm | % Polysperm |
|---|---|---|---|
| N2 | 53 | 0 | 0 (0/53) |
| marc-3(tm1626) I | 27 | 8 | 23 (8/35) |
| ubc-13(tm3546) IV | 38 | 0 | 0 (0/38) |
| marc-3(tm1626) I; ubc-13(tm3546) IV | 69 | 17 | 20 (17/86) |

**20 °C**

| RNAi | Monosperm | Polysperm | % Polysperm |
|---|---|---|---|
| rrf-1(pk1417) I; hTfR(RNAi) (F2) | 93 | 0 | 0 (0/93) |
| rrf-1(pk1417) I; marc-3(RNAi) (F2) | 68 | 7 | 9.3 (7/75) |
| rrf-1(pk1417) I; let-70(RNAi) (F1) | 200 | 5 | 2.4 (5/205) |

Polysperm phenotype of zygotes was scored under fluorescent microscopy after immunostaining respective zygotes with a monoclonal antibody, 1CB4, which recognizes sperm-derived organelles, MOs, with nuclear DNA counter staining. The sums of the numbers obtained from several experiments are shown.

identification of paternal pronuclei in zygotes, we treated these animals with *mat-1* RNAi from the L4 larval to adult stage to prevent the progression of the meiotic cell cycle of zygotes beyond metaphase I, as described previously[10]. We found a higher polysperm frequency in zygotes produced from the *marc-3(tm1626); egg-3(tm1191)* double homozygous mutant hermaphrodites (33.3%) than in zygotes produced from the *marc-3(tm1626)* homo; *egg-3(tm1191)* hetero siblings (19.4%) and zygotes from the *egg-3(tm1191)* single homozygous mutant hermaphrodites (16.4%, Table 2). These results suggest that MARC-3 and the EGG-complex component EGG-3 function additively in the polysperm block.

To determine the timing of the second sperm entry into the *marc-3* mutant zygotes after the first one, we performed a series of live imaging analyses using a *marc-3(tm1626)* mutant strain expressing sperm-specific *spe-11* promoter-driven HSP-6::mCherry and germline-specific *pie-1* promoter-driven GFP::PH(PLCδ1), which label sperm-derived mitochondria and oocyte-derived PM, respectively. In *C. elegans*, the oocyte changes its morphology to a round shape through cortical rearrangement during maturation, and then moves to the spermatheca for fertilization (Fig. 6A). Fertilized eggs exit from the spermatheca to the uterus within 5 min after fertilization and start embryogenesis (Fig. 6A). Chitin layer formation is considered to occur within 5 min after fertilization[8]. We observed the bispermy phenotype in 9 out of 38 specimens of *marc-3(tm1626)* mutant zygotes (24%) through the live imaging analysis. The timing of dual-sperm entry was recorded six times (Fig. 6B, C; Movie 2). Among these, the time interval between the first and the second sperm entry was within 12 s (0, 0, 1.5, 9, 12, and 12 s, mean ± S.D. = 5.75 s ± 5.88). Therefore, the second sperm appeared to enter the zygote either simultaneously or immediately after the first

sperm entry when the polysperm block was compromised in the *marc-3(tm1626)* mutant. We also examined the timing of polysperm occurred by the loss of EGG-3, the essential component of the EGG complex (Fig. 6B, C; Movie 3). In the *egg-3(tm1191)* mutant hermaphrodites expressing *spe-11* promoter-driven HSP-6::mCherry, the second sperm entry into the zygote was observed in 7 out of 25 specimens (28%). Whereas the *marc-3(tm1626)* mutant zygotes accepted the second sperm within 12 s after the first sperm entry, the second sperm entry into the *egg-3(tm1191)* zygotes was observed 198 to 238.5 s after the first sperm entry (198, 213, 222, and 238.5 s), approximately at the timing when these zygotes exited from the spermatheca to the uterus (Fig. 6B, C). These observations suggest that MARC-3 and EGG-3 conduct polysperm block sequentially, namely, the fast and the slow polysperm block, respectively, during fertilization.

**RING-CH domain of MARC-3 is critical to prevent polysperm**
We further examined whether the RING-CH domain of MARC-3 was critical for MARC-3 function. Since the substitution of the 38th cysteine to serine (C38S) in the RING-CH domain abrogated the in vitro auto-ubiquitination activity (Fig. 4C), we introduced this C38S substitution into MARC-3 (Fig. 7A) and examined whether it would bind to LET-70 in a yeast two-hybrid assay (Fig. 7B). The C38S-substituted MARC-3N lost its ability to bind to LET-70 (Fig. 7B), suggesting that the RING-CH domain is critical for their binding.

The polysperm phenotype of the *marc-3(tm1626)* mutant was completely rescued by introducing a transgene encoding the functional MARC-3::GFP (Table 2 and Fig. 7C, the left column). In contrast, when an equivalent MARC-3::GFP transgene with a RING-CH domain mutation (C38S) was introduced into the *marc-3(tm1626)* mutant (Supplementary Fig. 4A), the polyspermic phenotype was not rescued (Table 2 and Fig. 7C, the right column). Furthermore, the internalization of MARC-3(C38S)::GFP was delayed compared with that of MARC-3::GFP, and the protein harboring the mutation tended to remain on the PM in ovulated oocytes until they passed through the spermatheca in the *marc-3* mutant gonads (Fig. 7D, arrows). This suggests that MARC-3 itself is a substrate for ubiquitination. These results support the view that the RING-CH domain of MARC-3 is important for blocking polysperm and lysosomal degradation.

**MARC-3 may contribute to polysperm block in mature oocytes**
Although MARC-3 and the EGG complex seem to function independently in the fast and the slow polysperm block, respectively, we further investigated the functional relationship between MARC-3 and the EGG-complex components. We first examined their possible physical interactions using a yeast two-hybrid system (Fig. 7E). For this analysis, we used both the N-terminal and C-terminal cytoplasmic regions of MARC-3 (MARC-3N and MARC-3C) as baits. LacZ activation was detected only between MARC-3N and EGG-3 (Fig. 7E), and this activation was abrogated by C38S substitution into MARC-3N (Fig. 7E). These observations imply that the RING-CH domain of MARC-3 potentially binds to EGG-3, in addition to LET-70. AlphaFold2 predicted that MARC-3 might interact with EGG-3 and LET-70 in a similar manner (Supplementary Fig. 8A, B, respectively). We also compared the subcellular localization of MARC-3::GFP and mCherry::EGG-3 (Fig. 7F). MARC-3::GFP partially colocalized with mCherry::EGG-3 on the PM in maturing oocytes and was subsequently internalized from the PM, whereas mCherry::EGG-3 remained on the PM in ovulated oocytes. After fertilization, mCherry::EGG-3 was internalized from the PM, co-localized with MARC-3::GFP on endosomes, and delivered to lysosomes for degradation (Fig. 7F).

We also examined whether the loss of MARC-3 affected the dynamics of EGG-complex components during the oocyte-to-embryo transition. Reportedly, CHS-1 and EGG-3 interact with each other and are internalized and degraded in fertilized zygotes[8]. To measure the precise timing of their internalization, we introduced *spe-11* promotor-

**Table 2 | Polyspermy phenotype observed in *marc-3* mutants after various genetic manipulations**

| 20 °C | | | |
|---|---|---|---|
| **Strain** | **Monospermy** | **Polyspermy** | **% Polyspermy** |
| *marc-3(tm1626) I*, 6x backcrossed | 103 | 31 | 23 (31/134) |
| *marc-3(tm1626) I; Ppie-1::marc-3(WT)::GFP #1* | 70 | 0 | 0 (0/70) |
| *marc-3(tm1626) I; Ppie-1::marc-3(WT)::GFP #2* | 67 | 0 | 0 (0/67) |
| *marc-3(tm1626) I*, in which *Ppie-1::marc-3(WT)::GFP* was crossed out from the above strain | 78 | 13 | 14 (13/91) |
| *marc-3(tm1626) I; Ppie-1::marc-3(C38S)::GFP #1* | 74 | 16 | 18 (16/90) |
| **20 °C** | | | |
| **Mother** | **Monospermy** | **Polyspermy** | **% Polyspermy** |
| unmated *marc-3(tm1626)* hermaphrodites | 63 | 27 | 30 (27/90) |
| *marc-3(tm1626)* hermaphrodites mated with N2 males | 68 | 10 | 13 (10/78) |
| N2 hermaphrodites mated with *marc-3(tm1626)* males | 45 | 0 | 0 (0/45) |
| **20 °C** | | | |
| **Genotype** | **Monospermy** | **Polyspermy** | **% Polyspermy** |
| *egg-3(tm1191)/mIn1[mIs14 dpy-10(e128)] II, mat-1 RNAi* treated since the L4 stage | 157 | 0 | 0 (0/157) |
| *egg-3(tm1191) II, mat-1 RNAi* treated since the L4 stage | 92 | 18 | 16 (18/110) |
| *marc-3(tm1626) I; egg-3(tm1191)/mIn1[mIs14 dpy-10(e128)] II, mat-1 RNAi* treated since the L4 stage | 112 | 27 | 19 (27/139) |
| *marc-3(tm1626) I; egg-3(tm1191) II, mat-1 RNAi* treated since the L4 stage | 78 | 39 | 33 (39/117) |

Polyspermy phenotype was analyzed as described in Table 1. The sums of the numbers obtained from several experiments are shown. *egg-3(tm1191)* homozygous mutant hermaphrodites were the siblings of AD226: *egg-3(tm1191)/mIn1[mIs14 dpy-10(e128)]* heterozygous hermaphrodites and *marc-3(tm1626); egg-3(tm1191)* double homozygous mutant hermaphrodites were the siblings of GK2451: *marc-3(tm1626); egg-3(tm1191)/mIn1[mIs14 dpy-10(e128)]* hermaphrodites, in which *marc-3* was homozygous and *egg-3* was heterozygous. *mat-1* RNAi was treated to *egg-3* mutant-containing strains to arrest embryos before the onset of mitotic divisions. Otherwise, endomitotically duplicated chromosomes obscured the judgement of polyspermy phenotype in the *egg-3* mutant background, in which eggshell fails to form and shows severe polarity and osmolarity defective (Pod) phenotype.

driven HSP-6::mCherry and *pie-1* promotor-driven GFP::EGG-3 or GFP::CHS-1 into WT and *marc-3(tm1626)* mutant animals and performed time-lapse live imaging analyses (Supplementary Fig. 9A). We observed that GFP::EGG-3 was internalized on average at 15.9 min and 15.1 min after fertilization in WT ($n = 6$) and *marc-3(tm1626)* mutant zygotes ($n = 5$), respectively (Supplementary Fig. 9A). Therefore, the time interval between fertilization and EGG-3 internalization was not significantly different between the WT and the *marc-3(tm1626)* mutant (Supplementary Fig. 9A, Wilcoxon rank-sum test, $p > 0.70$). GFP::CHS-1 was also internalized at a statistically indistinguishable timing between the WT ($n = 6$) and *marc-3(tm1626)* mutant ($n = 4$) at approximately 15 min after fertilization (Supplementary Fig. 9A, Wilcoxon rank-sum test, $p > 0.40$). The internalization of EGG-3 and CHS-1 occurred much later than the timing of polyspermy observed in the *marc-3(tm1626)* mutant (Fig. 6B, C; less than 12 s after fertilization). Although the internalization timing of GFP::CHS-1 was superficially normal in the *marc-3(tm1626)* background (Supplementary Fig. 9A), its degradation was delayed in the *marc-3(tm1626)* mutant (Supplementary Fig. 2B), suggesting that MARC-3 affects the efficient delivery of internalized CHS-1 to the lysosomes after fertilization.

A previous study showed that RNAi of EGG-complex components, such as EGG-1 and EGG-2, caused polyspermy, deformation, and fragmentation of oocytes during ovulation[10]. Therefore, we examined whether mature *marc-3(tm1626)* oocytes were misshapen or fragmented before or during fertilization. We observed mature oocytes in WT and *marc-3(tm1626)* mutant adult hermaphrodites expressing *spe-11* promotor-driven HSP-6::mCherry and the PM marker *pie-1* promotor-driven GFP::PH(PLCδ1) (Supplementary Fig. 9B). Compared with the WT control, more oocyte fragments were observed in the *marc-3(tm1626)* mutant gonads (Supplementary Fig. 9B, $p < 0.007$, Wilcoxon rank-sum test). However, it should be noted that polyspermy zygotes were produced from both the fragmented oocytes and the unfragmented oocytes. That is, among 52 live-imaged oocytes, 4 out of 14 fragmented oocytes and 9 out of 38 unfragmented oocytes became polyspermy zygotes. This result implies that polyspermy is not necessarily caused by oocyte fragmentation and that there is no direct correlation between polyspermy and oocyte fragmentation.

## Discussion

In this study, we identified MARC-3 as an E3 ubiquitin ligase that selectively degrades maternal PM proteins after fertilization. In addition, we found that loss of MARC-3 resulted in a polyspermy phenotype in some zygote populations. These results suggest that MARC-3-mediated ubiquitination regulates both the fast polyspermy block during fertilization and the selective degradation of maternal membrane proteins after fertilization.

MARC-3 deficiency results in delayed degradation of maternal membrane proteins, such as CAV-1 and RME-2, after fertilization. Although K63-linked ubiquitin signals accumulated on endosomes in WT zygotes, they were attenuated in MARC-3-deficient zygotes, suggesting that MARC-3 acts on K63-linked polyubiquitination of maternal membrane proteins after fertilization. This phenotype was similar to that observed in the UBC-13 or UEV-1-deficient mutants[20]. Thus, MARC-3 may work with UBC-13 and UEV-1 in the K63-linked polyubiquitination of maternal PM proteins. However, their interaction was not detected in our yeast two-hybrid experiments. The presence of the three components may be necessary for their interaction in vivo. The delay in the degradation of maternal membrane proteins in *marc-3*-deficient embryos was milder than that observed in *ubc-13*- or *uev-1*-deficient embryos. Furthermore, K63-linked ubiquitin signals on endosomes were reduced but were not completely lost in *marc-3*-deficient embryos. Therefore, in addition to MARC-3, other E3 ubiquitin ligases may participate in this process.

In contrast, we identified LET-70/UBC-2 as an E2 ubiquitin-conjugating enzyme that interacts with MARC-3. LET-70 is the *C. elegans* homolog of mammalian UBE2D2 (UBC4, UbcH5b) or UBE2D3 (UbcH5c) and is widely involved in biological processes regulated by the ubiquitin-proteasomal system in *C. elegans*[30]. In fertilized eggs, LET-70 is known to cooperate with a RING-finger subunit of APC/C, APC-11, to exert ubiquitination activity for regulating the metaphase to anaphase transition during meiosis I[35]. Additionally, LET-70 is involved in the proteasomal degradation of maternal cytosolic proteins containing the CCCH finger domain in somatic cell lineages[36]. In this study, we showed that LET-70 is involved in the endocytic degradation of maternal PM proteins after fertilization in association with MARC-3.

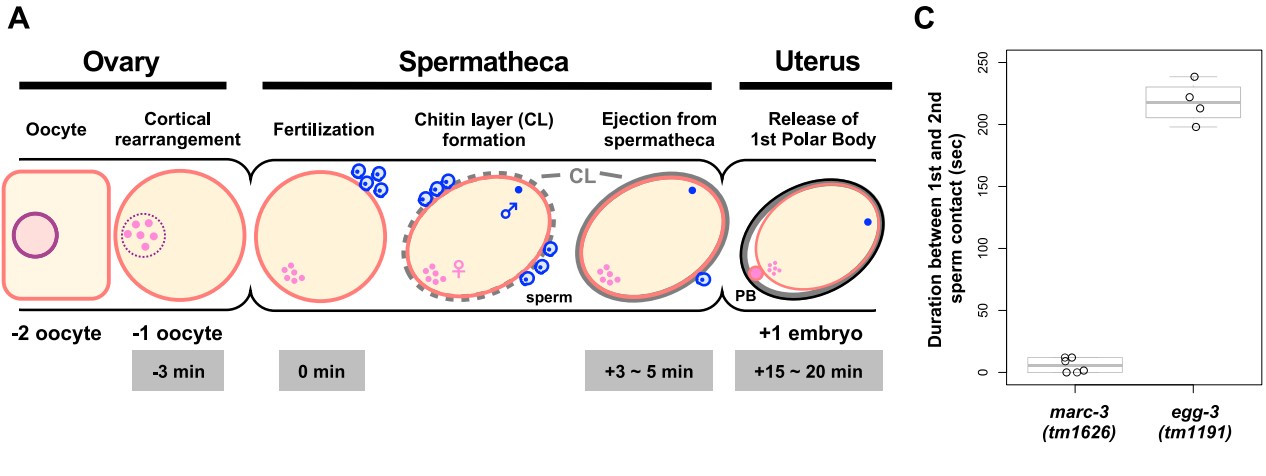

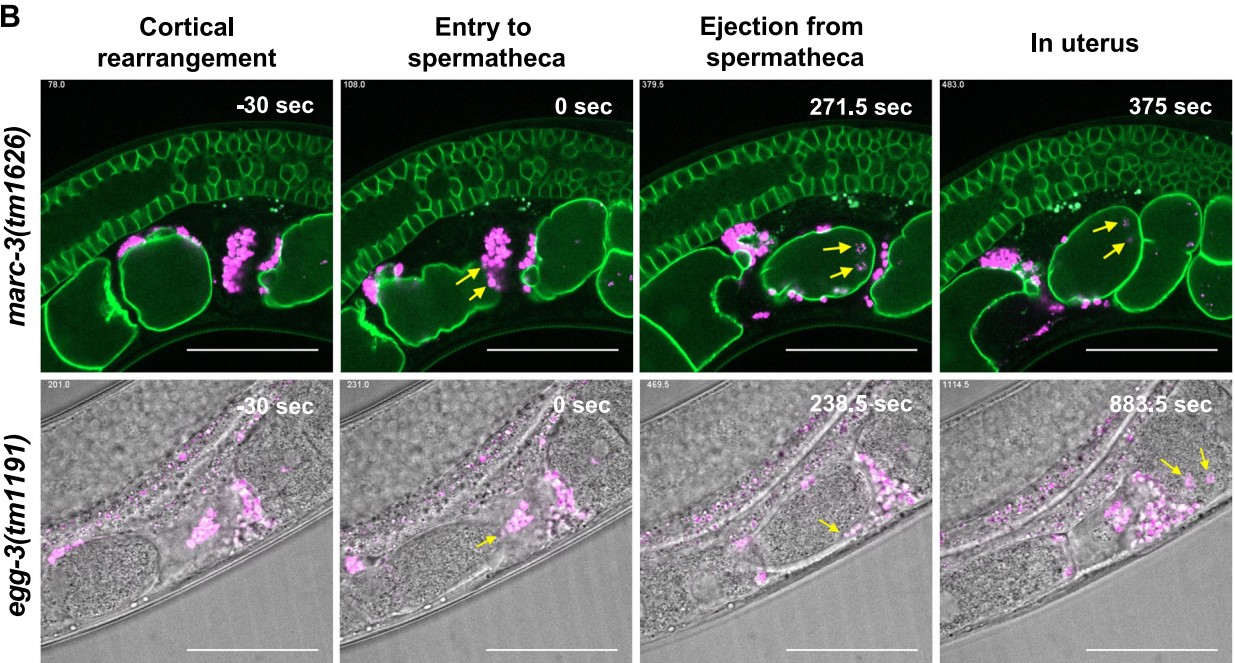

**Fig. 6 | Loss of *marc-3* and *egg-3* causes a defect in fast and slow polyspermy block, respectively. A** Schema of oocyte-to-embryo transition in *C. elegans* gonad. An oocyte at 3 min before fertilization changes its morphology through cortical rearrangement, then moves to the spermatheca for fertilization. The fertilized egg promptly forms a chitin layer (CL) and exits from the spermatheca to the uterus 3–5 min after fertilization. In this diagram, three oocytes are drawn in the spermatheca to illustrate the time course of their entry, fertilization, chitin layer formation, and exit from the spermatheca for ease of comprehension. **B** Live imaging of *marc-3* and *egg-3* mutant gonads displaying polyspermy. The second sperm entered nearly at the same timing as the first sperm in *marc-3(tm1626)* mutant (the second panel from the left in the upper row, arrows). In contrast, the second sperm entered the zygote more than 3 min after the first sperm entry in *egg-3(tm1191)* mutant (the second panel from the right in the lower row, an arrow). Eventually, these dual spermatozoa were observed inside of the polyspermy eggs in the uterus (the first and second panels from the right in the upper row, the rightmost panel in the lower row, arrows). Time (s) displayed on the panels indicates the duration from the first sperm contact (0 s). Magenta, sperm mitochondria; yellow arrows, sperm entered the zygote. Scale bars, 50 μm. **C** Distribution of durations between the first and the second sperm entry in *marc-3(tm1626)* and *egg-3(tm1191)* mutants. n = 6 for *marc-3(tm1626)* and n = 4 for *egg-3(tm1191)*. Boxplot: 25th to 75th percentiles with a line at the median; whiskers extend to 1.5 times the interquartile range; individual dots indicate respective durations.

We showed that MBP fusion with the RING-CH domain of MARC-3 exhibited ubiquitination activity in the presence of the mouse E1 enzyme and UbcH5a in vitro. Consistent with this result, it has been reported that GST fusion with the RING-CH domain of rat MARCH3 showed ubiquitination activity in the presence of E2 proteins, such as UbcH5c, UbcH6, and UbcH9, in vitro[25]. These observations support the hypothesis that MARC-3 functions with a UbcH5 homolog for ubiquitination in *C. elegans* embryos. MARCH family ubiquitin ligases were originally identified as homologs of the viral E3 ubiquitin ligases K3 and K5 derived from Kaposi's sarcoma-associated herpesvirus (KSHV)[37]. Reportedly, K3 ubiquitinates MHC class I molecules on the PM for internalization and lysosomal degradation[38,39]. During this process, K3 ubiquitin ligase mediates the mono-ubiquitination and K63-linked polyubiquitination in the presence of UbcH5b/c and Ubc13, respectively. In this scenario, MARC-3 may regulate mono-ubiquitination with LET-70 and polyubiquitination with UBC-13/UEV-1 for selective maternal PM protein degradation.

In addition to maternal membrane protein clearance, our results demonstrate that a MARC-3-mediated ubiquitination mechanism blocks polyspermy in *C. elegans*. The two phenotypes of *marc-3*, that is, delay of maternal membrane protein clearance and polyspermy, are likely to be independent because the delay of maternal membrane protein endocytosis (approximately 15 min after fertilization) occurred much later than the timing of polyspermy, which was not observed in

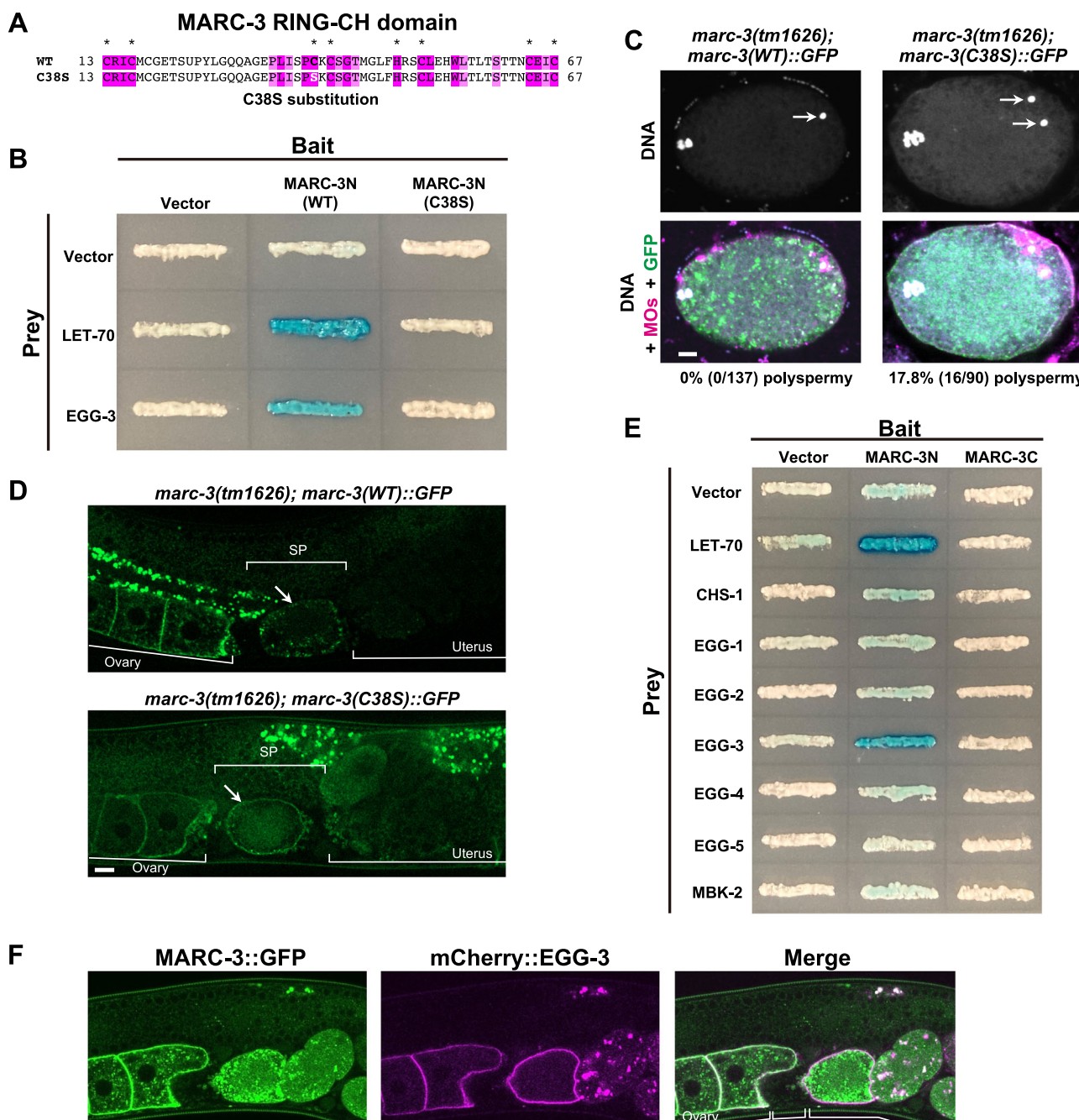

**Fig. 7 | RING-CH domain of MARC-3 is critical to prevent polyspermy. A** The RING-CH domain of MARC-3. Conserved amino acid residues among *C. elegans* MARC family proteins are marked with magenta-colored boxes. Among them, conserved cysteine and histidine residues are marked with asterisks. For the C38S substitution, the 38th cysteine is substituted with serine. See Supplementary Fig. 1B. **B** A yeast two-hybrid assay testing whether MARC-3N(C38S), in which the 38th cysteine was substituted with serine, can still bind to LET-70 and EGG-3 as WT MARC-3N. **C** WT *marc-3::GFP* transgene rescued (left, 0/137 zygotes were polyspermy), however, the C38S-substituted *marc-3::GFP* transgene failed to rescue (right, 16/90 zygotes were polyspermy) the polyspermy phenotype of *marc-3(tm1626)* mutant zygotes. Green, MARC-3::GFP. Magenta, MOs, White, nuclear DNA. Arrows indicate paternal pronuclei. Bar, 5 μm. Polyspermy frequencies in the respective zygotes are shown at the bottom. **D** Dynamic change in subcellular localization of MARC-3(WT)::GFP (upper panel) and MARC-3(C38S)::GFP (lower panel) in oocytes and early embryos in *marc-3(tm1626)* adult hermaphrodite gonads. Arrows, ovulated oocytes. Bar, 10 μm. SP, spermatheca. Compared with MARC-3(WT)::GFP, MARC-3(C38S)::GFP tended to remain on the PM of ovulated oocytes in *marc-3(tm1626)* mutant gonads (*n* = 11 for MARC-3(WT)::GFP, *n* = 24 for MARC-3(C38S)::GFP). **E** A yeast two-hybrid assay testing physical interactions between MARC-3 and the EGG-complex components. The assay used both the N-terminal and C-terminal cytoplasmic regions of MARC-3 (MARC-3N and MARC-3C) as the bait. The central largest cytoplasmic region of CHS-1, the N-terminal cytoplasmic regions of EGG-1 and EGG-2, and the full-length EGG-3, -4, -5, and MBK-2 were used as the prey. LET-70 was used as the positive-control prey. LacZ was used as the reporter. **F** Subcellular localization of MARC-3::GFP (green) and mCherry::EGG-3 (magenta) in oocytes and early embryos in an adult hermaphrodite gonad. Bar, 10 μm. SP, spermatheca. MARC-3::GFP partially colocalized with mCherry::EGG-3 (*n* = 8 hermaphrodites).

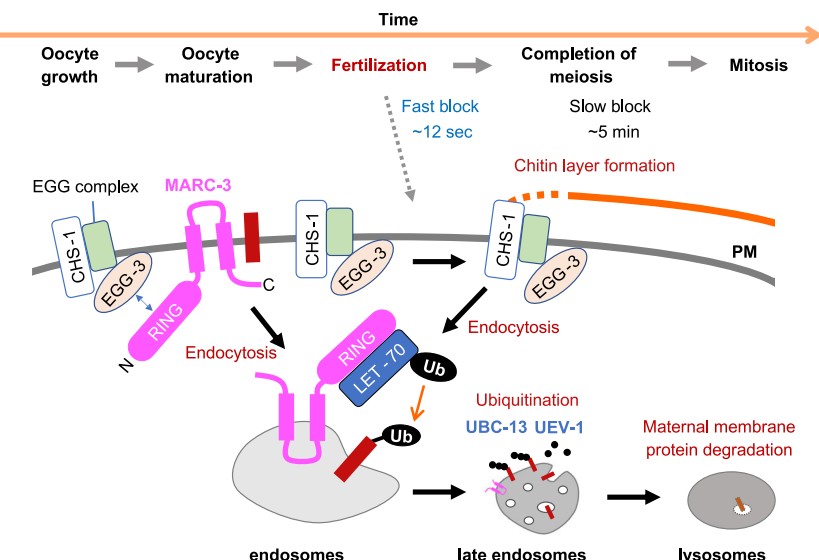

**Fig. 8 | Our working model for MARC-3 functions.** MARC-3 functions in the fast polyspermy block upon fertilization and in selective degradation of maternal membrane proteins after fertilization. MARC-3 localizes to the plasma membrane and punctate structures in growing oocytes and may interact with the EGG complex through binding to EGG-3. During oocyte maturation, MARC-3 is internalized from the PM and interacts with the E2 enzyme LET-70, triggering the fast polyspermy block. Some unidentified oocyte membrane proteins (brown bars) may be internalized from the PM and subjected to MARC-3-mediated ubiquitination before fertilization to prevent polyspermy. CHS-1 and the EGG complex then begin chitin layer formation for the slow polyspermy block. MARC-3 functions together with other E2 enzymes such as UBC-13 and UEV-1 in the selective degradation of maternal membrane proteins. RING: RING- CH domain; PM: plasma membrane; Ub: ubiquitin.

either *ubc-13-* or *uev-1-*deficient zygotes. Polyspermy block clearly depends on the ubiquitin ligase activity of MARC-3 and LET-70 appears to be an E2 partner of MARC-3. However, this conclusion is still arguable, because *let-70* RNAi caused polyspermy at a lower frequency compared with *marc-3* RNAi.

In *C. elegans*, chitin synthesis regulated by the EGG complex has been the sole known mechanism for polyspermy block. Eggshell chitin synthesis occurs within 5 min after fertilization[8,40]. Because oocytes ovulate into a spermatheca filled with many sperm cells, the so-called fast polyspermy block may also exist in addition to the chitin layer formation[8,10]. Our live-imaging analysis showed that polyspermy occurred within 0–12 s after fertilization in *marc-3(tm1626)* mutant zygotes. In contrast to the polyspermy observed in the *marc-3* mutant, the entry of the second sperm into the *egg-3(tm1191)* zygotes occurred more slowly, that is, at approximately 200–240 s (3–4 min) after the fertilization. Genetic analysis also revealed that the *marc-3*; *egg-3* double mutant caused a higher frequency of polyspermy than each single mutant. This result further supports the possibility that MARC-3 and EGG-3 function in different fashions to prevent polyspermy. Thus, MARC-3 and the EGG complex likely function in the fast and the slow polyspermy block, respectively (Fig. 8). The *egg-3* mutant received a second sperm 3–4 min after fertilization, approximately at the timing when the zygote exited from the spermatheca to the uterus, even though the MARC-3-mediated mechanism still existed. This observation suggests that the MARC-3-mediated fast polyspermy block is only temporarily effective, that is, only when the zygote resides within the spermatheca. Subsequently, eggshell chitin synthesis would be necessary for a permanent polyspermy block, such as the zona pellucida in other animals.

We also showed that the RING finger domain of MARC-3 is important for its binding to EGG-3 and LET-70 in a yeast two-hybrid assay. The current model for the MARC-3 functions is shown in Fig. 8. Since MARC-3 and EGG-3 showed similar subcellular localization pattern in growing oocytes, these proteins may transiently interact and execute a cooperative function in growing oocytes. In maturing oocytes, MARC-3 is internalized from the PM before ovulation, whereas EGG-3 remains on the PM. MARC-3 may dissociate from EGG-3 and bind to LET-70 to drive ubiquitination, block polyspermy, and degrade maternal membrane proteins. After fertilization, MARC-3 may function together with UBC-13 and UEV-1 for selective degradation of maternal membrane proteins, including MARC-3 itself, in embryos.

To summarize, we showed that MARC-3, a membrane-bound ubiquitin ligase, is required for the degradation of a subset of maternal PM proteins and for the fast polyspermy block upon fertilization. Our results also suggest that the ubiquitination mechanism is involved in the spatiotemporal changes in oocyte membrane properties during the oocyte-to-embryo transition and in the contact, recognition, and reaction to a single sperm upon fertilization. Future analyses of eggs lacking MARCH3 homologs in other organisms would clarify whether these mechanisms are universally conserved in sexually reproducing organisms.

## Methods

### General methods and generation of *marc-2*, *marc-3* mutant strains, and *marc-3::GFP* transgenic animals

*C. elegans* strains were handled and cultured as described previously[41]. All primers and *C. elegans* strains used in this study are listed in Supplementary Table 2 and 3, respectively. A CRISPR-Cas9 genome-edited allele of *marc-2*, *syb3694*, was provided by SunyBiotech (https://www.sunybiotech.com/). *syb3694* was detected by BspHI digestion of single-worm PCR products generated using the primers *marc-2(syb3694)*IL and *marc-2(syb3694)*IR (see the primer list for their sequences). Dr. Shohei Mitani generated and provided a deletion allele of *marc-3/C17E4.3*, *tm1626*, at the Japanese National Bioresource Project for the Experimental Animal "Nematode *C. elegans*". To detect the *tm1626* deletion mutation, single-worm PCR was conducted using primers C17E4.3delF2 and C17E4.3delR1 (see the primer list for the sequence). Another loss-of-function *marc-3* allele, *syb5493*, generated by CRISPR-Cas9 genome editing, was provided by SunyBiotech. The *syb5493* mutation was detected by *Avr*II digestion of single-worm PCR products generated using the primers C17E4.3delF2 and C17E4.3delR1. For phenotypic analysis, *marc-3(tm1626)* was outcrossed six times with N2. An endogenous GFP knock-in allele of *marc-3*, *syb8421*, was generated using CRISPR-Cas9 genome editing by inserting a DNA fragment

encoding GFP before the stop codon of the *marc-3* gene to drive the endogenous expression of C-terminally GFP-tagged MARC-3. All experimental protocols were approved by the Gunma University Genetic Modification Safety Committee (approval no. 22–006).

## Construction of plasmids and transgenes

Plasmid DNA was constructed for gene expression in *C. elegans*, yeast, and *Escherichia coli* using a Gateway Recombination Cloning System (Thermo Fisher Scientific). A genomic DNA fragment encompassing the entire *marc-3* coding sequence (1974 bp) was PCR-amplified using the primers C17E4.3-GWF2 and C17E4.3-GWR2 and cloned into pDONR221. The DNA fragment encoding the GFP sequence with flanking *Bam*HI sites was PCR-amplified from pDONR221-*cav-1::GFP* and inserted into an artificially created *Bam*HI site immediately before the *marc-3* stop codon. The resulting *marc-3::GFP* DNA fragment in pDONR221-*marc-3::GFP* was transferred to pID2.02 using the LR reaction for germline expression under the *pie-1* promoter. The substitution of the 38th cysteine to serine in the MARC-3 RING-CH domain was performed using the Q5 site-directed mutagenesis kit (New England Biolabs) using pDONR221-*marc-3::GFP* as the template and *marc-3*-C38S-F and *marc-3*-C38S-R as primers. Transgenic lines were generated using the microparticle bombardment method, as described previously[42]. Transgenes used in this study include *pwIs281[unc-119(+), Ppie-1::cav-1::GFP][5]*, *pwIs116[unc-119(+), Ppie-1::rme-2::GFP][43]*, *dkIs241[unc-119(+), Ppie-1::GFP::chs-1][20]*, *dkIs405[unc-119(+), Ppie-1::marc-3::GFP]* (this study), *pwIs20[unc-119(+), Ppie-1::GFP::rab-5][44]*, *pwIs40[unc-119(+), Ppie-1::mRFP::rab-7][20]*, *pwIs403[unc-119(+), Ppie-1::mCherry::rab-5]* (the Caenorhabditis Genetic Center), *dkIs698[unc-119(+), Pspe-11::hsp-6::mCherry][28]*, *ltIs38[unc-119(+), Ppie-1::GFP::PH(PLC1delta1)][45]*, *asIs2[unc-119(+), Ppie-1::mCherry::egg-3][8]*, *nnIs2[unc-119(+) Ppie-1::GFP::chs-1][8]*, and *dkIs1092[unc-119(+), Ppie-1::marc-3(C38S)::GFP]* (this study).

## RNAi experiments

The feeding RNAi method was used to knockdown gene expression[46]. Feeding RNAi bacterial clones, including those of *marc-1, -2, -3*, and *-6*, were obtained from the Ahringer genomic RNAi library[47]. A few cDNAs, including those of *marc-4*, and *marc-5* were PCR-amplified from a cDNA library and cloned into the L4440 vector using the Gateway system (Thermo Fisher Scientific) and the primers listed in the primer list (Supplementary Table 2). For the feeding RNAi method, the *E. coli* strain HT115(DE3) carrying an L4440 vector was spotted on NGM agar plates containing 5 mM isopropyl β-D-thiogalactopyranoside (IPTG) and incubated at 37 °C overnight to express the double-strand RNA of the gene to be downregulated. Next, L4-larval-stage hermaphrodites of an appropriate genetic background were transferred onto the feeding RNAi plates and incubated for an appropriate period at 20 °C. The resulting phenotype was examined in the transferred hermaphrodites and their progenies. An L4440 vector containing cDNA of the human transferrin receptor (*hTfR*) gene was used as a negative control.

## Microscopy and immunostaining

Living transgenic worms expressing GFP, mRFP, and mCherry were placed in a drop of M9 buffer containing levamisole on agarose pad of a glass slide and covered with a coverslip. The worms on the glass slides were observed with an Olympus FV1000 or FV1200 confocal microscope equipped with a 60×, 1.35 NA UPlanSApo or a 100×, 1.40 NA UPlanSApo oil objective lens (Olympus Corp., Tokyo, Japan), and visualized with a software FV10-ASW v4.2.3.6. For immunostaining, young adult hermaphrodites were dissected in a drop of M9 buffer on an adhesive glass slide, and extruded gonads and embryos were freeze-cracked and fixed with cold methanol and acetone[17]. After blocking with PTA (phosphate-buffered saline containing 3% bovine serum albumin, 0.1% Tween-20, 0.05% NaN₃, and 1 mM EDTA) or PTB (phosphate-buffered saline containing 1% bovine serum albumin, 0.1% Tween-20, 0.05% NaN₃, and 1 mM EDTA), the specimens were immunostained with the following primary antibodies: a mouse anti-ubiquitin antibody (FK2; Medical and Biological Laboratories, Nagoya, Japan, Cat. No. D058-3, dilution 1:10,000), rabbit anti-K48- and K63-linked ubiquitin chain antibodies (Apu2 and Apu3, respectively, Millipore, Tokyo, Japan: Apu2, Cat. No. 05-1307, diluted to 1:1000; Apu3, Cat. No. 05-1308, dilution 1:200), a mouse monoclonal anti-MO antibody (1CB4, a gift from Dr. S. L'Hernault, dilution 1:1000), a mouse anti-GFP monoclonal antibody (3E6, Life Technologies, Cat. No. A-11120, dilution 1:100), and a rabbit anti-MARC-3 polyclonal antibody (this study, dilution 1:50–100). Alexa Fluor 488 and 555 goat anti-mouse IgG (Life Technologies, Cat. No. A11029 and A21424, dilution 1:1000) and Alexa Fluor 488 and 555 goat anti-rabbit IgG (Life Technologies, Cat. No. A11034 and A21429, dilution 1:1000) were used as secondary antibodies. The specimens were counterstained with either 0.5 µg/ml DAPI or 20 µM DRAQ5 (BioStatus, Cat. No. DR50200) to stain the nuclear DNA. A rhodamine-conjugated chitin-binding probe (New England Biolabs, Cat. No. P5210S, dilution 1:500) was applied to fixed 1-cell embryos to detect the eggshell chitin layer. A lipophilic dye, FM4-64 (Invitrogen, Cat. No. F34653), was applied to unfixed embryos at 1 mM to test the permeability of eggshells formed around the embryos.

Rabbit anti-MARC-3 polyclonal antibody was generated by TK Craft Corporation against a peptide corresponding to the MARC-3 N-terminal region (PYLGQQAGEPLISPC; OPERON Biotechnologies) and affinity-purified with the same peptide. The antibody was preabsorbed in *marc-3(tm1626)* worm powder before use. The specificity of these antibodies was verified by immunostaining the wild-type and *marc-3(tm1626)* strains (Supplementary Fig. 6B). It should be noted that this antibody showed some non-specific signals in somatic spermathecal cells. Images of immunostaining and live imaging, which were taken by an Olympus FV1000 or FV1200 confocal microscope, were digitally quantified using ImageJ2 (version: 2.14.0/1.54f).

## Live imaging

L4-stage hermaphrodites were placed on a seeded NGM plate and incubated overnight at either 20 °C. After incubation, two to seven young adult hermaphrodites were transferred to a drop of M9 buffer containing levamisole (0.5 mM) on a 10% agarose pad (Nacalai Tesque), which was created inside an annular silicone sheet placed on a glass slide. The transferred worms were enclosed and immobilized on a coverslip. Time-lapse imaging was performed using a CSU-W1 spinning-disc confocal system (Yokogawa Electric Corporation) attached to an inverted microscope IX71 (Olympus) and an sCMOS camera Sona (Andor). The glass slide containing the specimen was held on a constant-temperature unit ThermoPlate (Tokai Hit) and maintained at 20 °C. The exposure settings and interval for photography were 500 ms with 20% laser power at 470 nm excitation, 500 ms with 80% laser power at 555 nm excitation, and 150 ms with DIC. Visualization was performed with a software Micro-Manager v2.0.0. Movies were created using ImageJ (version: 1.53k) from the sequential images obtained.

## Cell fractionation

The fractionation assay was performed as described previously[48]. Briefly, young adult animals were collected from NGM plates using M9 buffer and washed several times with M9 buffer. We suspended them in a fractionation buffer (20 mM Hepes/KOH (pH 7.4), 10 mM KCl, 1.5 mM MgCl₂, 1 mM EDTA, 1 mM EGTA, 250 mM sucrose), in which 1 mM dithiothreitol, protease inhibitors (Complete EDTA-free, Roche Diagnostics GmbH), and 0.1 mM phenylmethylsulfonyl fluoride (PMSF) were added before homogenization. The samples were then homogenized in an ice-cold Dounce stainless-steel tissue grinder (Wheaton). The homogenates were centrifuged at $750 \times g$ for 5 min at 4 °C to remove nuclear fraction and debris. The resulting supernatant was further centrifuged at $100,000 \times g$ for 1 hr at 4 °C. Goat anti-GFP

antibody (Fitzgerald Industries International, Cat. No. 70R-GG001, diluted to 1:2000) was used to detect MARC-3::GFP. Rabbit anti-CCT-5 antibody[49], diluted to 1:500, and mouse monoclonal anti-LMP-1 antibody (DSHB, Iowa City, IA, USA, diluted to 1:500) were used as markers for the cytosolic and membrane fractions, respectively. As secondary antibodies, donkey anti-goat IgG, HRP conjugate (Millipore, Cat. No. AP180P), peroxidase-conjugated goat anti-rabbit IgG (Jackson ImmunoResearch, Cat. No. 111-035-003), and peroxidase-conjugated goat anti-mouse IgG (Jackson ImmunoResearch, Cat. No. 115-035-003), respectively, were used at 1:5,000 dilution. Uncropped full-scan images of western blots used for making Fig. 3A were shown in Supplementary Fig. 10.

### In vitro ubiquitination assay

To generate recombinant proteins containing the N-terminal RING-finger domain of MARC-3 and its mutant form, the DNA fragment encoding amino acids 2–101 of MARC-3 was PCR-amplified using the primers mch3RINGFBamHI and mch3RINGRPstI from a *C. elegans* cDNA library (DupLEX-A; OriGene Technologies). The PCR product was digested with *Bam*HI and *Pst*I and inserted into the multiple pMAL-c2X (NEB) cloning sites to generate pMAL-c2X MCH-3RING. The DNA fragment encoding an N-terminal RING-finger domain, in which the 38th cysteine residue in the domain was substituted with serine, was generated by PCR-mediated site-directed mutagenesis using pMAL-c2X MCH-3RING as the template in combination with the point-mutation-containing primers mch3C38SF and mch3C38SR. The maltose-binding protein (MBP)-tagged MARC-3 RING finger domain and its mutant form were cloned into the pMAL-c2X vector and expressed in the *E. coli* BL21(DE3) codon-plus RIL strain (Stratagene). Bacterial cells expressing MBP-tagged proteins were lysed in column buffer (20 mM Tris-HCl, pH 7.5, 200 mM NaCl, 1 mM DTT, 100 μM $ZnSO_4$, 1 mg/mL lysozyme (Wako), and a protease inhibitor cocktail (Roche)) on ice, sonicated, and centrifuged at $5800 \times g$ for 10 min. The supernatant was filtered and loaded onto an Amylose Resin column (Amersham GE Healthcare) on ice overnight and washed with column buffer. The fusion protein was eluted from the column using a column buffer containing 10 mM maltose.

The in vitro ubiquitination assay was performed as described previously[50]. The purified MBP-MARC-3 RING-finger domain and its mutant form (20 μg/mL) were incubated in a reaction buffer (50 mM Tris-HCl, pH 8.8, 2 mM DTT, 5 mM $MgCl_2$, and 4 mM ATP) with 50 μg/mL of ubiquitin (Sigma), 1.6 μg/mL of recombinant mouse E1, and 20 μg/mL of purified UbcH5a at 32 °C for 2 h. The reaction products were subjected to immunoblotting using either rabbit anti-MBP polyclonal antibody (New England Biolabs, Cat. No. E8030S, dilution 1:4,000) or mouse anti-ubiquitin monoclonal antibody (P4D1, Santa Cruz Biotechnology, Cat. No. sc-8017, dilution 1:500) as the primary antibody and either goat anti-rabbit IgG-alkaline phosphatase (Santa Cruz Biotechnology, Cat. No. sc-2007, dilution 1:5000) or goat anti-mouse IgG-alkaline phosphatase (Santa Cruz Biotechnology, Cat. No. sc-2008, dilution 1:10,000) as the secondary antibody.

### Yeast two-hybrid assay

The yeast two-hybrid assay used the DupLEX-A Yeast Two-Hybrid System (OriGene Technologies) as previously described[51]. cDNA fragments encoding MARC-3 N-terminal cytoplasmic region (MARC-3N, 1–98 aa), MARC-3 C-terminal cytoplasmic region (MARC-3C, 179–431 aa), LET-70 (full length, 1–147 aa), UBC-13 (full length, 1–151 aa), UEV-1 (full length, 1–139 aa), CHS-1 (the largest central cytoplasmic region, 418–836 aa), EGG-1 (the N-terminal cytoplasmic region, 1–48 aa), EGG-2 (the N-terminal cytoplasmic region, 1–49 aa), EGG-3 (full length, 1–555 aa), EGG-4 (full length, 1–753 aa), EGG-5 (full length, 1–753 aa), and MBK-2 (full length isoform a, 1–508 aa) were PCR amplified from a DupLEX *C. elegans* cDNA library using Q5 high-fidelity DNA polymerase (NEB) using the primer pairs *marc-3N*-GWF3 and *marc-3N*-GWR3 for

MARC-3N, C17E4.3-GWF3 and C17E4.3-GWR1 for MARC-3C, *let-70*-GWF and *let-70*-GWR for LET-70, *ubc-13*-GWF and *ubc-13*cDNA-GWR for UBC-13, *uev-1*-GWF and *uev-1*-GWR for UEV-1, *chs-1*(418–836)-GWF1 and *chs-1*(418–836)-GWR1 for CHS-1, *egg-1*(1–48)-GWF1 and *egg-1*(1–48)-GWR1 for EGG-1, *egg-2*(1–49)-GWF1 and *egg-2*(1–49)-GWR1 for EGG-2, *egg-3*-GWF1 and *egg-3*-GWR1 for EGG-3, *egg-4/5*-GWF1 and *egg-4*-GWR2 for EGG-4, *egg-4/5*-GWF1 and *egg-5*-GWR2 for EGG-5, and *mbk-2*-GWF1 and *mbk-2*-GWR1 for MBK-2. The C38S-substituted *marc-3N* cDNA fragment was PCR-amplified from pMAL-c2X MCH-3RING(C38S) using primers *marc-3N*-GWF3 and *marc-3N*-GWR3. The cDNA fragments were cloned into the bait vector pEG202 or prey vector pJG4-5 using the Gateway recombination cloning system (Invitrogen Life Technologies). Briefly, the respective cDNA fragments were first cloned into the pDNOR221 donor vector by the BP reaction and then transferred to destination vectors, such as the bait plasmid pEG202Gtwy or prey plasmid pJG4-5Gtwy, by the LR reaction. The resulting bait and prey plasmids were introduced into the yeast EGY48 strain containing the lacZ reporter pSH18-34 plasmid. Transformants were incubated on X-gal (5-bromo-4-chloro-3-indolyl-β-D-galactopyranoside) plates to examine the physical interactions among these proteins in yeast cells[36].

### AlphaFold2 prediction

The 3D-docking structures of EGG-3, LET-70, and MARC-3 were predicted by ColabFold v1.5.3 with a default setting using "alphafold2_multimer_v3" model type[52,53]. Visualization was performed using ChimeraX v1.7[54–56].

### Statistical analysis

Multiple unpaired *t*-tests (two-tailed) were performed using the Prism 9 software (GraphPad ver. 9.5.1). The Wilcoxon rank-sum test was performed using the R software (ver. 3.6.1) using the "wilcox.exact" function of the "exactRankTests" library. No statistical method was used to predetermine sample size. No data were excluded from the analyses. The experiments were not randomized. The Investigators were not blinded to allocation during experiments and outcome assessment.

### Reporting summary

Further information on research design is available in the Nature Portfolio Reporting Summary linked to this article.

## Data availability

The data supporting these findings are available from the authors upon request. Source data for graphs (Figs. 1C, D, E, F, 2B, 6C) and blots (Fig. 3A) are provided as a Source data file and Supplementary Fig. 10. Source data are provided with this paper.

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

## Acknowledgements

We thank Drs. Barth D. Grant, Steven L'Hernault, Anjon Audhya, and Shohei Mitani for supplying reagents. Many strains used in this study were provided by the Caenorhabditis Genetic Center, funded by the NIH Office of Research Infrastructure Programs (P40 OD010440), the National Bioresource Project for the Experimental Animal "Nematode *C. elegans*". The anti-LMP-1 monoclonal antibody was obtained from the Developmental Studies Hybridoma Bank, created by the NICHD of the NIH and maintained at The University of Iowa. We thank Dr. Yuhkoh Satouh and all members of the K. Sato and M. Sato Laboratories for their kind assistance and helpful discussion. This study was supported by the Japan Society for the Promotion of Science KAKENHI (grant numbers 19H05711 and 20H00466 to Ken Sato (K.S.), 23KJ0291 to Kenta Sugiura (KT.S.), 22K15097 to Taeko Sasaki, 19H05712 to Noriyuki Matsuda (N.M.) and 19H05712 and 21H02472 to Miyuki Sato (M.S.)). This study was also supported by the joint research program of Institute for Molecular and Cellular Regulation, Gunma University (23009 to N.M. and M.S.) and Nanken-Kyoten (2023-kokunai11 to N.M. and M.S.) of Tokyo Medical and Dental University.

## Author contributions

K.S., M.S., I.K., KT.S., and N.M. designed the experiments. K.S. and M.S. identified MARC-3, which is involved in selective degradation of maternal membrane proteins. I.K., KT.S., M.S., T.S., and K.S. characterized the *marc-3*-deficient animals. N.M. performed in vitro ubiquitination assay. K.S. and M.S. supervised the project. K.S., M.S., I.K., KT.S., and N.M. wrote the manuscript.

## Competing interests

The authors declare no competing interests.
