## [Peer Review File · Nature Communications]

MARC-3, a membrane-associated ubiquitin ligase, is required for fast polyspermy block in *Caenorhabditis elegans*REVIEWER COMMENTS

Reviewer #1 (Remarks to the Author):

Comments:

In this manuscript, Kawasaki, Sugiura, and colleagues report that mutations in a membrane-associated ubiquitin ligase, MARC-3, cause an incompletely penetrant phenotype in which multiple sperm fertilize the oocyte. They show that K63-linked ubiquitin conjugates associated with endosomes are diminished in *marc-3* mutant early embryos. Consistent with this result, several oocyte membrane proteins persist in *marc-3* mutant embryos, suggesting that MARC-3 may directly promote their degradation. Much of the data presented is of high quality and the finding that a membrane-associated ubiquitin ligase functions to prevent polyspermy is of interest to the field. Yet, there are several weaknesses in the manuscript that require attention before this work can be published. The principal weaknesses are: (1) the *marc-3* mutant phenotype, its expression pattern, and its membrane association, should be more completely described; and (2) the relationship between EGG-3 and MARC-3 remains murky in the current treatment.

(1) The *marc-3* mutant phenotype and related issues. The authors show that *marc-3* mutants exhibit an incompletely penetrant polyspermic phenotype with approximately 18–30% of embryos analyzed showing this phenotype. From this description, I would assume that a homozygous mutant strain is viable and fertile and can be maintained as a stock. The authors should conduct brood-size measurements of the wild-type and mutants. The authors should comment on whether any additional phenotypes are observed. The authors report the “MARC-3 expression pattern” using a transgene in which a promoter from another gene (*pie-1*) drives MARC-3::GFP expression. Given the ease of genome editing, the standard in the field is to examine an endogenously tagged protein. This should be done. From the data presented, tagging with GFP at the MARC-3 C-terminus is likely to be a fruitful approach. Finally, the authors should conduct some simple biochemical fractionation experiments to show that MARC-3 is indeed membrane associated.

(2) The relationship between MARC-3 and EGG-3. It is not clear what the authors are trying to conclude from the data presented in the second half of Table 1 in which they conduct RNA interference (RNAi) with *egg-3* and *chs-1* in the wild-type and *marc-3* mutant backgrounds? Are the authors wanting to conclude that *marc-3* and *egg-3* function in a common pathway? If so, then these experiments need to be repeated using mutants instead of RNAi. One notable difference between *marc-3* mutants and *egg-3* mutants is that the latter, but not the former, is defective in eggshell synthesis. Perhaps this observation, taken together with the observation that polyspermy can be observed within 10 seconds of fertilization in *marc-3* mutants, has led to the suggestion that *marc-3* mutations affect a “fast block” to polyspermy. I think it would be important for the authors to examine *egg-3* mutants and see whether they observe a similar or different timing of polyspermy as compared to *marc-3* mutants. Is there evidence that these proteins interact *in vivo*?

(3) Oocyte fragments. In the Discussion, the authors speculate that MARC-3 may function prior to fertilization to control monospermy because they observe oocyte fragments in the gonads of *marc-3* mutants. These oocyte fragments undoubtedly arise from defective ovulation in which the spermatheca closes and constricts a portion of the oocyte before the ovulatory process is completed. This could easily be verified by time-lapse recordings. That defects in ovulation might be observed in *marc-3* mutants is not surprising because the

authors show that several oocyte membrane proteins are affected in the mutants. Since the maturing oocyte signals its own ovulation, this effect may have nothing to do with the polyspermic phenotype. Can the authors comment on whether they only observe polyspermy if a fragment pinches off the oocyte or whether these aspects of the phenotype are independent as expected. As an aside, the images shown in Extended Data Figure 2 seem to indicate that RME-2::GFP (and possibly CAV-1::GFP) might be more abundant in oocytes in the mutant (it would be good to extend the fluorescence intensity analysis oocytes in addition to 1–4-cell embryos).

David Greenstein

Reviewer #2 (Remarks to the Author):

Fertilization in *C. elegans* involves ovulation into a chamber (the spermatheca) full of fertilization-competent spermatozoa. Essentially, every oocyte gets fertilized under these conditions, an eggshell principally composed of chitin forms and the resulting diploid cells commence embryogenesis. Polyspermy is almost never observed in wild type hermaphrodite self-fertilization and, while there have been a few insights into egg membrane proteins that seem to be involved, much still remains to be learned about how this species avoids that outcome. This study makes significant progress in addressing that important question as it identifies a ubiquitin E3 ligase, MARC-3, that is required to facilitate the *C. elegans* analog of the classic “fast block” to polyspermy. Among the affected substrates, are several oocytes plasma membrane transmembrane (TM) proteins previously implicated in blocking polyspermy, indicating that the ubiquitin-mediated clearance of these proteins phenocopies loss of function of these TM proteins. These are, so-far, unique aspects to how a lack of MARC-3 allows polyspermy to occur. Unlike prior analyses of oocyte TM proteins where their loss of function causes polyspermy, loss of *marc-3* function does not appear to affect formation of the chitin eggshell. This is an important result because it rules out the eggshell as the reason for the “fast block” equivalent in *C. elegans*. This manuscript also identifies the E2 ligase as *let-70*, which is not what was anticipated from the author’s prior work. So, this work has identified a previously unknown player in the fast block to polyspermy that has a unique phenotype. Furthermore, this work also nicely shows that there must be at least two non-redundant parallel pathways involving ubiquitin-mediated protein degradation operative during the oocyte-to-zygote transition. All in all, this reviewer has considerable enthusiasm for the work described in this manuscript. The work is interesting, the manuscript is well-written and the conclusions are well-supported and conservative. As such, this reviewer strongly recommends publication in *Nature Communications*.

Minor criticisms

On the left panel of Figure 1B, indicate the positions of the ovary, spermatheca and uterus, as is already the case for the right panel of Figure 1B. The same issue applies to extended data Figures 2A and B.

Lines 345-347: Figure 6D, indicate oocytes and fertilized eggs on the figure and the zygote where internalization has occurred. Yes, this is obvious, but only to expert readers accustomed to looking at such a figure.

Reviewer #3 (Remarks to the Author):

In this very interesting manuscript, Kawasaki and colleagues investigate the molecular basis of the selective degradation of maternal plasma membrane proteins that takes place after fertilization. This process is part of the oocyte-to-zygote transition, during which maternal factors required for early embryogenesis are progressively degraded in the fertilized egg as zygotic expression takes over.

Building upon previous results that implicated ubiquitination and, more specifically, the UBC-13/UEV-1 E2 complex into the endocytosis and endosome degradation of maternal plasma membrane proteins, Kawasaki et al. use *C. elegans* as a model system to identify MARC-3 as a membrane-associated E3 ubiquitin ligase that is required for endosomal K63-linked ubiquitination of a subset of maternal membrane proteins in zygotes. Consistent with such a function, MARC-3 shows a dynamic subcellular localization by being largely localized to the plasma membrane in growing oocytes, undergoing endocytosis in ovulated and fertilized oocytes, and finally being degraded during early embryogenesis. The authors also show that MARC-3 interacts with LET-70, a ubiquitin-conjugating enzyme, via its N-terminal cytoplasmic RING-CH-type domain; moreover, the latter was found to have auto-ubiquitination activity *in vitro* in the presence of Ubch5a, a human ortholog of LET-70. This suggests that MARC-3 and LET-70 could form an active E2-E3 complex, independently of UBC-13/UEV-1 (which did not appear to interact with MARC-3 in yeast two-hybrid experiments).

Remarkably, by further studying the *marc-3* mutant, Kawasaki et al. also found that up to 30% of *marc-3* zygotes displayed a polyspermy phenotype that could be rescued by a MARC-3::GFP transgene. Live imaging of fertilization suggests that MARC-3 contributes to a fast block to polyspermy that is established within seconds of the first gamete fusion event, consistent with the observation that the *marc-3* mutation does not affect the formation of the chitin layer (whose synthesis takes minutes and is responsible for the slow block to polyspermy in *C. elegans*). Finally, the authors show that the polyspermy-blocking function of MARC-3 also depends on its RING-CH domain (which in this case could possibly act independently of LET-70), and that - similarly to what was previously observed by RNAi depletion of *egg-1/2* - loss of the protein may facilitate polyspermy by reducing the integrity of the oocyte's cortex.

This is a carefully executed and well written study that not only brings an important contribution to our understanding of how maternal membrane proteins are selectively degraded after fertilization, but also suggests that a (EGG-3?) ubiquitination-mediated fast block to polyspermy exists in *C. elegans*. Considering the evolutionary conservation of MARC-3, this work could also have important implications for the same processes in mammals. I only have a few minor comments that the authors could consider while revising their manuscript:

- 1) The sentences within the Abstract use different tenses, but for consistency I would suggest to stick to present tense.
- 2) Although I understand the historical development of the study, which first addresses maternal protein degradation in the zygote and then stumbles onto the block to polyspermy, I wonder whether, within the Introduction (but not the results, whose order I would not change), these two topics could not be better presented in the order in which they occur in nature. In the current version, the paragraphs that describe them are anyway essentially independent from each other, and one of the key points of the work is precisely the identification of MERC-3 as a factor that connects them, by being implicated in both.

3) Perhaps I have missed this, but what was the reason for using Ubch5a as a proxy for LET-70 in in vitro ubiquitination assays, rather than LET-70 itself?

4) Have the authors considered modelling the MARC-3 RING-CH domain/LET-70 complex with AlphaFold-Multimer, in order to bring further support to their experimental results?

5) Compatibly with space constraints, I think that a final figure that schematically summarizes the two functions of MERC-3 (with possible interactors) would help readers remember the take-home messages of this beautiful study.

Response to Reviewers

Reviewer #1:

In this manuscript, Kawasaki, Sugiura, and colleagues report that mutations in a membrane-associated ubiquitin ligase, MARC-3, cause an incompletely penetrant phenotype in which multiple sperm fertilize the oocyte. They show that K63-linked ubiquitin conjugates associated with endosomes are diminished in *marc-3* mutant early embryos. Consistent with this result, several oocyte membrane proteins persist in *marc-3* mutant embryos, suggesting that MARC-3 may directly promote their degradation. Much of the data presented is of high quality and the finding that a membrane-associated ubiquitin ligase functions to prevent polyspermy is of interest to the field. Yet, there are several weaknesses in the manuscript that require attention before this work can be published. The principal weaknesses are: (1) the *marc-3* mutant phenotype, its expression pattern, and its membrane association, should be more completely described; and (2) the relationship between EGG-3 and MARC-3 remains murky in the current treatment.

(1) The *marc-3* mutant phenotype and related issues. The authors show that *marc-3* mutants exhibit an incompletely penetrant polyspermic phenotype with approximately 18–30% of embryos analyzed showing this phenotype. From this description, I would assume that a homozygous mutant strain is viable and fertile and can be maintained as a stock. The authors should conduct brood-size measurements of the wild-type and mutants. The authors should comment on whether any additional phenotypes are observed. The authors report the “MARC-3 expression pattern” using a transgene in which a promoter from another gene (*pie-1*) drives MARC-3::GFP expression. Given the ease of genome editing, the standard in the field is to examine an endogenously tagged protein. This should be done. From the data presented, tagging with GFP at the MARC-3 C-terminus is likely to be a fruitful approach. Finally, the authors should conduct some simple biochemical fractionation experiments to show that MARC-3 is indeed membrane associated.

Response: We appreciate the reviewer’s comments. Based on this comment, we examined the brood sizes of the wild-type N2 and *marc-3* null mutant strains (Table 1). We found that *marc-3* null mutant animals exhibited a reduced progeny size compared with the wild-type animals (248.1 ± 18.1 (n=10); 80.7% of N2). In addition, we observed that 11% of the eggs laid were arrested during embryogenesis, and 3.5% were arrested after

hatching. Therefore, we believe that polyspermy in *marc-3* mutants is pathogenic. However, because the total developmental arrest frequency of the *marc-3* mutant (14.5%) was slightly lower than the observed frequency of polyspermy (18-30%), we are currently not sure whether 100% of *marc-3* polyspermy embryos are arrested during development. Further studies are required to elucidate the time course and mechanism by which *marc-3* polyspermy induces developmental arrest.

In accordance with the reviewer's comment, we also examined the localization and dynamics of endogenous MARC-3 during the oocyte-to-embryo transition. First, as suggested, we generated an endogenous *marc-3::GFP* transgenic strain using the CRISPR-Cas9 genome editing method. As the reviewer mentioned, a DNA fragment encoding GFP was inserted at the MARC-3 C-terminus before the *marc-3* stop codon. We confirmed that MARC-3::GFP, which was expressed under its own promoter, showed very similar dynamics and expression patterns to those driven by a germline-specific *pie-1* promoter (Fig. S6A). Second, we generated an antibody against MARC-3 and confirmed that endogenous MARC-3 was predominantly expressed in oocytes and early embryos and that MARC-3 was internalized during ovulation and degraded in lysosomes after fertilization, as observed using the *pie-1*-promotor-driven *marc-3::GFP* transgene (Fig. S6B).

Furthermore, we conducted a cell fractionation experiment to examine the membrane association of MARC-3::GFP and confirmed that MARC-3::GFP was reproducibly fractionated into the membrane fraction (n=3), indicating that it was indeed a transmembrane protein (Fig. 3A).

(2) The relationship between MARC-3 and EGG-3. It is not clear what the authors are trying to conclude from the data presented in the second half of Table 1 in which they conduct RNA interference (RNAi) with *egg-3* and *chs-1* in the wild-type and *marc-3* mutant backgrounds? Are the authors wanting to conclude that *marc-3* and *egg-3* function in a common pathway? If so, then these experiments need to be repeated using mutants instead of RNAi. One notable difference between *marc-3* mutants and *egg-3* mutants is that the latter, but not the former, is defective in eggshell synthesis. Perhaps this observation, taken together with the observation that polyspermy can be observed within 10 seconds of fertilization in *marc-3* mutants, has led to the suggestion that *marc-3* mutations affect a "fast block" to polyspermy. I think it would be important for the authors to examine *egg-3* mutants and see whether they observe a similar or different timing of polyspermy as compared to *marc-3* mutants. Is there evidence that these proteins interact *in vivo*?

Response: Thank you for the suggestion. In accordance with the reviewer's comments, we have further examined the relationship between EGG-3 and MARC-3 using an *egg-3* knockout strain instead of RNAi. We generated the GK2451: *marc-3(tm1626) I; egg-3(tm1191)/mIn1[mIs14, dpy-10] II* balanced viable double mutant strain and compared the polyspermy frequencies between its *marc-3; egg-3* double homozygous mutant progeny and the *marc-3; egg-3/mIn1* single homozygous mutant progeny (*marc-3* homo but *egg-3* hetero), along with the polyspermy frequencies of *egg-3* single homozygous mutant progeny and *egg-3/mIn1* heterozygous mutant progeny produced from the AD226: *egg-3(tm1191)/mIn1[mIs14, dpy-10] II* balanced *egg-3* mutant strain. We found that the *marc-3; egg-3* double homozygous mutant hermaphrodites showed a higher frequency of polyspermy (33.3%) than the *marc-3* homo; *egg-3* hetero-siblings (19.4%) and *egg-3* single homozygous mutant hermaphrodites (16.4%) (see revised Table 2).

Furthermore, through live imaging analysis of the polyspermy timing of *marc-3* and *egg-3* mutants, we found that while the *marc-3* mutant zygote accepted the second sperm within several seconds after the first sperm entry, second sperm acceptance into the *egg-3* mutant zygote occurred approximately 200 s after the first sperm entry (See new Fig. 6). These observations suggest that MARC-3 and EGG-3 function separately in fast and slow polyspermy blocks, respectively, during fertilization. Based on these results, we would like to correct our previous conclusion that *marc-3* and *egg-3* function via a common pathway. We now consider that *marc-3* and *egg-3* function in different pathways for the polyspermy block. We have included these new data and revised the manuscript accordingly. In particular, some parts of the revised manuscript, including the Discussion section, have been improved based on the reviewer's insightful suggestions.

Based on the reviewer's comment, we have examined the possible physical interactions between MARC-3 and EGG-3 *in vivo* using a co-immunoprecipitation experiment. Unfortunately, it was difficult to clearly detect such an interaction because MARC-3 and EGG-3 are transmembrane and peripheral membrane proteins, respectively, and their interaction is assumed to be transient. We also solubilized the lysate with detergents, which may have dissociated these proteins. We have revised our manuscript to describe the functional relationship between MARC-3 and EGG-3.

(3) Oocyte fragments. In the Discussion, the authors speculate that MARC-3 may function prior to fertilization to control monospermy because they observe oocyte fragments in the gonads of *marc-3* mutants. These oocyte fragments undoubtedly arise

from defective ovulation in which the spermatheca closes and constricts a portion of the oocyte before the ovulatory process is completed. This could easily be verified by time-lapse recordings. That defects in ovulation might be observed in *marc-3* mutants is not surprising because the authors show that several oocyte membrane proteins are affected in the mutants. Since the maturing oocyte signals its own ovulation, this effect may have nothing to do with the polyspermic phenotype. Can the authors comment on whether they only observe polyspermy if a fragment pinches off the oocyte or whether these aspects of the phenotype are independent as expected. As an aside, the images shown in Extended Data Figure 2 seem to indicate that RME-2::GFP (and possibly CAV-1::GFP) might be more abundant in oocytes in the mutant (it would be good to extend the fluorescence intensity analysis oocytes in addition to 1–4-cell embryos).

Response: Thank you for the suggestion. In accordance with the reviewer's comment, we conducted live imaging experiments and carefully analyzed oocyte fragmentation and polyspermy phenotypes. We found that although the fragmentation was observed in 14 oocytes out of the total 52 live-imaged oocytes, polyspermy eggs were eventually identified from both the fragmented oocytes (4 out of 14) and the remaining unfragmented oocytes (9 out of 38, See the last section of Results, Fig. S9). These observations suggested that the polyspermy phenotype of *marc-3* null mutant does not necessarily correlate with the pinching-off phenotype of oocytes during ovulation. We speculate that MARC-3 as well as some subunits of the EGG complex (EGG-1 and EGG-2) may help ensure the integrity of the oocyte cell surface, although the fragmentation phenotype and polyspermy could not be directly correlated. This point has also been mentioned in the revised Discussion.

In accordance with the reviewer's comment, we also examined the signal intensities of the maternal membrane proteins CAV-1::GFP, RME-2::GFP, and GFP::CHS-1 in -3 to -1 oocytes, in addition to 1-, 2-, and 4-cell-stage embryos (Fig. 1C and Fig. S2A and B). We confirmed that in contrast to early embryos, the signal intensities of these GFP-tagged maternal membrane proteins did not significantly differ between wild-type and *marc-3* mutant oocytes. Therefore, we concluded that the increase in maternal membrane protein signals in early embryos of *marc-3* mutant was not due to increased production or accumulation of these maternal proteins in their oocytes.

Reviewer #2:

Fertilization in *C. elegans* involves ovulation into a chamber (the spermatheca) full of

fertilization-competent spermatozoa. Essentially, every oocyte gets fertilized under these conditions, an eggshell principally composed of chitin forms and the resulting diploid cells commence embryogenesis. Polyspermy is almost never observed in wild-type hermaphrodite self-fertilization and, while there have been a few insights into egg membrane proteins that seem to be involved, much still remains to be learned about how this species avoids that outcome. This study makes significant progress in addressing that important question as it identifies a ubiquitin E3 ligase, MARC-3, that is required to facilitate the *C. elegans* analog of the classic “fast block” to polyspermy. Among the affected substrates, are several oocytes plasma membrane transmembrane (TM) proteins previously implicated in blocking polyspermy, indicating that the ubiquitin-mediated clearance of these proteins phenocopies loss of function of these TM proteins. These are, so-far, unique aspects to how a lack of MARC-3 allows polyspermy to occur. Unlike prior analyses of oocyte TM proteins where their loss of function causes polyspermy, loss of *marc-3* function does not appear to affect formation of the chitin eggshell. This is an important result because it rules out the eggshell as the reason for the “fast block” equivalent in *C. elegans*. This manuscript also identifies the E2 ligase as *let-70*, which is not what was anticipated from the author’s prior work. So, this work has identified a previously unknown player in the fast block to polyspermy that has a unique phenotype. Furthermore, this work also nicely shows that there must be at least two non-redundant parallel pathways involving ubiquitin-mediated protein degradation operative during the oocyte-to-zygote transition. All in all, this reviewer has considerable enthusiasm for the work described in this manuscript. The work is interesting, the manuscript is well-written, and the conclusions are well-supported and conservative. As such, this reviewer strongly recommends publication in *Nature Communications*.

Minor criticisms

1) On the left panel of Figure 1B, indicate the positions of the ovary, spermatheca and uterus, as is already the case for the right panel of Figure 1B. The same issue applies to extended data Figures 2A and B.

Response: Thank you for the suggestion. Based on the reviewer’s comments, we have added this information to the left image panel of revised Fig. 1C and the gonad images in the revised Fig. S2A and B.

2) Lines 345-347: Figure 6D, indicate oocytes and fertilized eggs on the figure and the zygote where internalization has occurred. Yes, this is obvious, but only to expert readers

accustomed to looking at such a figure.

Response: Since we have added a new Fig. 6 to the revised manuscript, Fig. 6 in the original manuscript was moved to Fig.7. According to the suggestion, we have included positional information of the ovary, spermatheca, and uterus in the images of revised Fig. 7D and 7F (Fig. 6F and Fig. 6D, respectively, in the original manuscript) to better understand manuscript description as added to the revised Fig. 1C and the Fig. S2A and B.

Reviewer #3:

In this very interesting manuscript, Kawasaki and colleagues investigate the molecular basis of the selective degradation of maternal plasma membrane proteins that takes place after fertilization. This process is part of the oocyte-to-zygote transition, during which maternal factors required for early embryogenesis are progressively degraded in the fertilized egg as zygotic expression takes over.

Building upon previous results that implicated ubiquitination and, more specifically, the UBC-13/UEV-1 E2 complex into the endocytosis and endosome degradation of maternal plasma membrane proteins, Kawasaki *et al.* use *C. elegans* as a model system to identify MARC-3 as a membrane-associated E3 ubiquitin ligase that is required for endosomal K63-linked ubiquitination of a subset of maternal membrane proteins in zygotes. Consistent with such a function, MARC-3 shows a dynamic subcellular localization by being largely localized to the plasma membrane in growing oocytes, undergoing endocytosis in ovulated and fertilized oocytes, and finally being degraded during early embryogenesis. The authors also show that MARC-3 interacts with LET-70, a ubiquitin-conjugating enzyme, via its N-terminal cytoplasmic RING-CH-type domain; moreover, the latter was found to have auto-ubiquitination activity *in vitro* in the presence of UbcH5a, a human ortholog of LET-70. This suggests that MARC-3 and LET-70 could form an active E2-E3 complex, independently of UBC-13/UEV-1 (which did not appear to interact with MARC-3 in yeast two-hybrid experiments).

Remarkably, by further studying the *marc-3* mutant, Kawasaki *et al.* also found that up to 30% of *marc-3* zygotes displayed a polyspermy phenotype that could be rescued by a MARC-3::GFP transgene. Live imaging of fertilization suggests that MARC-3 contributes to a fast block to polyspermy that is established within seconds of the first gamete fusion event, consistent with the observation that the *marc-3* mutation does not affect the formation of the chitin layer (whose synthesis takes minutes and is responsible

for the slow block to polyspermy in *C. elegans*). Finally, the authors show that the polyspermy-blocking function of MARC-3 also depends on its RING-CH domain (which in this case could possibly act independently of LET-70), and that - similarly to what was previously observed by RNAi depletion of *egg-1/2* - loss of the protein may facilitate polyspermy by reducing the integrity of the oocyte's cortex.

This is a carefully executed and well written study that not only brings an important contribution to our understanding of how maternal membrane proteins are selectively degraded after fertilization, but also suggests that a (EGG-3?) ubiquitination-mediated fast block to polyspermy exists in *C. elegans*. Considering the evolutionary conservation of MARC-3, this work could also have important implications for the same processes in mammals. I only have a few minor comments that the authors could consider while revising their manuscript:

1) The sentences within the Abstract use different tenses, but for consistency I would suggest sticking to present tense.

Response: Thank you for the suggestion. According to the reviewer's comment, we have corrected the Abstract using the present tense.

2) Although I understand the historical development of the study, which first addresses maternal protein degradation in the zygote and then stumbles onto the block to polyspermy, I wonder whether, within the Introduction (but not the results, whose order I would not change), these two topics could not be better presented in the order in which they occur in nature. In the current version, the paragraphs that describe them are anyway essentially independent from each other, and one of the key points of the work is precisely the identification of MARC-3 as a factor that connects them, by being implicated in both.

Response: In accordance with the reviewer's comment, we have improved the Introduction by changing the order of the explanations.

3) Perhaps I have missed this, but what was the reason for using Ubch5a as a proxy for LET-70 in *in vitro* ubiquitination assays, rather than LET-70 itself?

Response: We appreciate the reviewer's comments. Ubch5a was selected because it is 94% identical to LET-70 in its amino acid sequence and has almost the same structure as that predicted by AlphaFold2 (See the revised Fig. S7A-C). Moreover,

UbcH5a has been generally used in in vitro ubiquitination assay as a representative of Ubc4/5 family E2 enzymes (Matsuda JCS 2001). In fact, the RING-finger domain of MARC-3 shows strong ubiquitination activity when UbcH5a is used as an E2 enzyme.

4) Have the authors considered modelling the MARC-3 RING-CH domain/LET-70 complex with AlphaFold-Multimer, in order to bring further support to their experimental results?

Response: In accordance with the reviewer's comment, we created a model representing the MARC-3 RING-CH domain/LET-70 complex using AlphaFold-Multimer software. We added this model to new Fig. S7D & E.

5) Compatibly with space constraints, I think that a final figure that schematically summarizes the two functions of MARC-3 (with possible interactors) would help readers remember the take-home messages of this beautiful study.

Response: According to the reviewer's comment, we have created a figure representing our current model of MARC-3 function during the oocyte-to-embryo transition, which has been added in Fig. 8.

REVIEWERS' COMMENTS

Reviewer #1 (Remarks to the Author):

The authors have done a superb job in revising the manuscript. This work is important because it: (1) identifies a membrane-associated ubiquitin ligase needed to promote the fast block to polyspermy; (2) shows that EGG-3 promotes a slow block to polyspermy; and (3) provides evidence that the fast and slow blocks to polyspermy act successively to ensure the formation of the diploid zygote. The cell biological, biochemical, and genetic data are of high quality and the results are convincing. Defining the relevant MARC-3 substrates will be an important next step. One exciting possibility is that a membrane protein needs to be turned over to install the fast block, and this must happen within seconds after fertilization. Alternatively, the MARC-3 could function before fertilization in the oocyte as the authors also discuss.

The authors should be commended on their fine work and their responsiveness to the prior critiques. The only very minor quibble I have is that I am not convinced that the authors need to evoke a model in which the oocyte cortex might be more fragile in *marc-3* mutants to account for the observed ovulation defect (e.g., oocyte fragmentation). The fragmentation undoubtedly occurs when the distal constriction of the spermatheca closes on the oocyte before it has fully entered the spermatheca (this can happen in many other genetic backgrounds that perturb ovulation). To conclude there is a membrane fragility phenotype, biophysical measurements would be needed (again this is very minor).

David Greenstein

Reviewer #2 (Remarks to the Author):

This is a re-review of a manuscript I thought was very nicely done the first time. I have looked over all comments by the three reviewers of the first version and I find all questions look like they have been satisfactorily answered. Consequently, I remain enthusiastic about this work and feel it is a very thorough study worthy of publication in Nature Communications.

One minor criticism:

In Figure 8, the endosome has a dark blue Ub tagged protein. It is unclear to me, and not explained in the figure legend, how this connects to other elements of the summary drawing. Is the blue tone of the color off (too dark?), does it relate to a protein that had previously been on the cell surface (if that's the case, put a similar structure on the surface on the left side of the drawing and use an arrow to indicate it is internalized?). One is left to wonder what the authors are trying to say.

Reviewer #3 (Remarks to the Author):

The authors essentially addressed the points that I raised in my previous review. I only have a few minor comments in relation with the new Supplementary Fig. 7:

1) In lines 240-241 it is stated that "UbcH5a is 94% identical to LET-70 in its amino acid sequence and has a similar 3D structure according to the AlphaFold2 prediction (Supplementary Fig. 7A-C)". But with such high sequence identity, one does not need an

AlphaFold prediction to conclude that that the two proteins will have a similar 3D structure!
Panels A-C of Supplementary Fig. 7 could thus in principle be omitted.

2) The title of Supplementary Fig. 7 is not completely correct and should thus be modified to "AlphaFold predictions of LET-70, UbcH5a and complexes of the MARC- 3 RING-CH domain with LET-70 or EGG-3" (or simply "AlphaFold predictions of the MARC- 3 RING-CH domain in complex with LET-70 or EGG-3", if the authors will decide to omit panels A-C from the figure). The following sentence ("3D structures of LET-70, UbcH5a, and possible binding between EGG-3 and the RING-CH domain of MARC-3 were simulated using AlphaFold2.") could also be deleted since it is redundant.

3) If panels A-C are kept, I would change "merge" to "superposition" in "(A-C) 3D structure predictions of LET-70 (A, green), UbcH5a (B, cyan), and their merge (C, green and cyan)."

4) In both panels D and E, I would replace "Binding model" to "AlphaFold prediction" and modify "with" to "in complex with".

5) It is difficult to judge things from a single view, but the predicted interfaces between MARC-3 RING CH and LET-70/EGG-3 do not seem to be very extensive (particularly in the case of LET-70). Although these of course remain just predictions, it would be informative if the authors specified the ipTM+pTM and ipTH scores of the two models in the corresponding figure legends.

Response to the referee's comments

Reviewer #1 (Remarks to the Author):

The authors have done a superb job in revising the manuscript. This work is important because it: (1) identifies a membrane-associated ubiquitin ligase needed to promote the fast block to polyspermy; (2) shows that EGG-3 promotes a slow block to polyspermy; and (3) provides evidence that the fast and slow blocks to polyspermy act successively to ensure the formation of the diploid zygote. The cell biological, biochemical, and genetic data are of high quality and the results are convincing. Defining the relevant MARC-3 substrates will be an important next step. One exciting possibility is that a membrane protein needs to be turned over to install the fast block, and this must happen within seconds after fertilization. Alternatively, the MARC-3 could function before fertilization in the oocyte as the authors also discuss.

1) The authors should be commended on their fine work and their responsiveness to the prior critiques. The only very minor quibble I have is that I am not convinced that the authors need to evoke a model in which the oocyte cortex might be more fragile in *marc-3* mutants to account for the observed ovulation defect (e.g., oocyte fragmentation). The fragmentation undoubtedly occurs when the distal constriction of the spermatheca closes on the oocyte before it has fully entered the spermatheca (this can happen in many other genetic backgrounds that perturb ovulation). To conclude there is a membrane fragility phenotype, biophysical measurements would be needed (again this is very minor)

We agree with the referee's comment. We have deleted the following sentences (yellow highlighting) from the result and discussion sections and just described the phenotype.

p13 line 29~ Results section in the previous version

This result suggests that the oocyte cortex of the *marc-3* mutant is slightly more fragile than that of WT, as observed after RNAi depletion of *egg-1/2*¹⁰, which may weaken the polyspermy block.

P16 line 11~ Discussion section in the previous version

Mature oocytes occasionally showed a partial cell fragmentation phenotype during ovulation in *marc-3*-deficient animals, as observed in oocytes lacking subunits of the EGG complex (EGG-1 or EGG-2) or CBD-1^{7, 10, 41}. This partial cell fragmentation phenotype has not

been reported in embryos depleted of CHS-1 or GNA-2, suggesting that this phenotype is independent from chitin synthesis after fertilization. Using yeast two-hybrid analysis, we found that MARC-3 potentially binds to EGG-3. One possibility might be that MARC-3 interacts with the EGG complex through EGG-3 in oocytes to ensure the integrity of the oocyte cell surface. However, it should be noted that the oocyte fragmentation phenotype and polyspermy appeared not directly correlated because polyspermy was observed both in fragmented and in unfragmented eggs⁴¹.

Reviewer #2 (Remarks to the Author):

This is a re-review of a manuscript I thought was very nicely done the first time. I have looked over all comments by the three reviewers of the first version and I find all questions look like they have been satisfactorily answered. Consequently, I remain enthusiastic about this work and feel it is a very thorough study worthy of publication in Nature Communications.

One minor criticism:

In Figure 8, the endosome has a dark blue Ub tagged protein. It is unclear to me, and not explained in the figure legend, how this connects to other elements of the summary drawing. Is the blue tone of the color off (too dark?), does it relate to a protein that had previously been on the cell surface (if that's the case, put a similar structure on the surface on the left side of the drawing and use an arrow to indicate it is internalized?). One is left to wonder what the authors are trying to say.

We appreciate the referee's suggestion. We intended that a dark blue Ub tagged protein in Fig.8 indicates an unidentified possible substrate of MARC-3-mediated ubiquitination, which is involved in polyspermy block. According the referee's suggestion, we have improved Fig.8 by adding the same structure on the cell surface on the left side of the drawing and described the explanation in more detail in the Figure legend.

Revised legend of Fig. 8

MARC-3 functions in the fast polyspermy block upon fertilization and selective degradation of maternal membrane proteins after fertilization. MARC-3 localizes to the plasma membrane and punctate structures in growing oocytes and may interact with the EGG complex through binding to EGG-3. During oocyte maturation, MARC-3 is internalized from the PM and interacts with the E2 enzyme LET-70, triggering the fast polyspermy block. Some unidentified oocyte membrane proteins (brown bars) may be internalized from the PM as well and subjected to MARC-3-mediated ubiquitination before fertilization to prevent the polyspermy. CHS-1 and the EGG

complex then begin chitin layer formation for the slow polyspermy block. MARC-3 functions together with other E2 enzymes such as UBC-13 and UEV-1 in the selective degradation of maternal membrane proteins. RING: RING- CH domain; PM: plasma membrane; Ub: ubiquitin.

Reviewer #3 (Remarks to the Author):

The authors essentially addressed the points that I raised in my previous review. I only have a few minor comments in relation with the new Supplementary Fig. 7:

1) In lines 240-241 it is stated that "UbcH5a is 94% identical to LET-70 in its amino acid sequence and has a similar 3D structure according to the AlphaFold2 prediction (Supplementary Fig. 7A-C)". But with such high sequence identity, one does not need an AlphaFold prediction to conclude that that the two proteins will have a similar 3D structure! Panels A-C of Supplementary Fig. 7 could thus in principle be omitted.

We agree with the referee's comment. According to the referee's comment we have omitted panel A-C from previous version of Supplementary Fig. 7 and moved to Supplementary Fig. 8 in the revised manuscript.

2) The title of Supplementary Fig. 7 is not completely correct and should thus be modified to "AlphaFold predictions of LET-70, UbcH5a and complexes of the MARC- 3 RING-CH domain with LET-70 or EGG-3" (or simply "AlphaFold predictions of the MARC- 3 RING-CH domain in complex with LET-70 or EGG-3", if the authors will decide to omit panels A-C from the figure). The following sentence ("3D structures of LET-70, UbcH5a, and possible binding between EGG-3 and the RING-CH domain of MARC-3 were simulated using AlphaFold2.") could also be deleted since it is redundant.

We appreciate the referee's suggestion. We have revised the legend of Supplementary Figure 8 as follows.

AlphaFold predictions of MARC-3 in complex with LET-70 or EGG-3

(A) AlphaFold2 prediction of MARC-3 (green) in complex with LET-70 (cyan). pLDDT=59.6, pTM=0.514, ipTM=0.861 (pTM+ipTM=1.375).

(B) AlphaFold2 prediction of MARC-3 (green) in complex with EGG-3 (magenta). pLDDT=57.1, pTM=0.544, ipTM=0.442 (pTM+ipTM=0.986).

The draft data and log files are available at

https://github.com/kentasugiura/kawasakietal_natcomm_rawdata/tree/main/AF2

3) If panels A-C are kept, I would change "merge" to "superposition" in "(A-C) 3D structure predictions of LET-70 (A, green), UbcH5a (B, cyan), and their merge (C, green and cyan)."

We have deleted panels A-C according to the referee's comment.

4) In both panels D and E, I would replace "Binding model" to "AlphaFold prediction" and modify "with" to "in complex with".

According to the referee's comment, we have corrected these points as described above.

5) It is difficult to judge things from a single view, but the predicted interfaces between MARC-3 RING CH and LET-70/EGG-3 do not seem to be very extensive (particularly in the case of LET-70). Although these of course remain just predictions, it would be informative if the authors specified the ipTM+pTM and ipTM scores of the two models in the corresponding figure legends.

We have added the ipTM+pTM and ipTM scores of the two models according to the referee's comment as described above.